# Think or Not? Exploring Thinking Efficiency in Large Reasoning Models via an Information-Theoretic Lens

**Xixian Yong**[1]    **Xiao Zhou**[123*]    **Yingying Zhang**[4]    **Jinlin Li**[1]
**Yefeng Zheng**[5]    **Xian Wu**[4*]

[1]Gaoling School of Artificial Intelligence, Renmin University of China
[2]Beijing Key Laboratory of Research on Large Models and Intelligent Governance
[3]Engineering Research Center of Next-Generation Intelligent Search and Recommendation, MOE
[4]Tencent Jarvis Lab    [5]Medical Artificial Intelligence Lab, Westlake University
{xixianyong, xiaozhou}@ruc.edu.cn   kevinxwu@tencent.com

## Abstract

The recent rise of Large Reasoning Models (LRMs) has significantly improved multi-step reasoning performance, but often at the cost of generating excessively long reasoning chains. This paper revisits the efficiency of such reasoning processes through an information-theoretic lens, revealing a fundamental trade-off between reasoning length and semantic efficiency. We propose two metrics—*InfoBias* and *InfoGain*—to quantify divergence from ideal reasoning paths and stepwise information contribution, respectively. Empirical analyses show that longer reasoning chains tend to exhibit higher information bias and diminishing information gain, especially for incorrect answers. Motivated by these findings, we introduce an **entropy-based Adaptive Think** strategy that dynamically halts reasoning once confidence is sufficiently high, improving efficiency while maintaining competitive accuracy. Compared to the Vanilla Think approach (default mode), our strategy yields a 1.10% improvement in average accuracy and a 50.80% reduction in token usage on QwQ-32B across six benchmark tasks spanning diverse reasoning types and difficulty levels, demonstrating superior efficiency and reasoning performance. These results underscore the promise of entropy-based methods for enhancing both accuracy and cost-effiiciency in large language model deployment. Code and data are available at https://github.com/chicosirius/think-or-not.

## 1   Introduction

With the paradigm of Large Language Models (LLMs) [Brown et al., 2020] extending from training-time scaling [Kaplan et al., 2020] to test-time scaling [Muennighoff et al., 2025], the emergence of Large Reasoning Models (LRMs) [Li et al., 2025]—such as OpenAI's o1 [OpenAI, 2024], Deepseek's R1 [Guo et al., 2025a], and QwQ-32B [Team, 2025b]—has significantly advanced the frontier of model reasoning capabilities. However, we observe a noteworthy trend: in pursuit of better performance, these models increasingly rely on lengthy Chain-of-Thought (CoT) [Wei et al., 2022] reasoning, leading to quadratic growth in computational complexity. This prolonged internal or external "deep thinking" process contradicts the principle of cognitive economy observed in human reasoning, thereby undermining the efficiency of LRMs in practical applications [Su et al., 2025].

Inspired by Shannon's three-level model of communication [Shannon, 1948], we revisit the phenomenon of excessively long reasoning chains in contemporary LRMs. **At the technical level**, extending the reasoning chain can be interpreted as injecting redundant bits into a noisy channel to

---

*Corresponding authors.

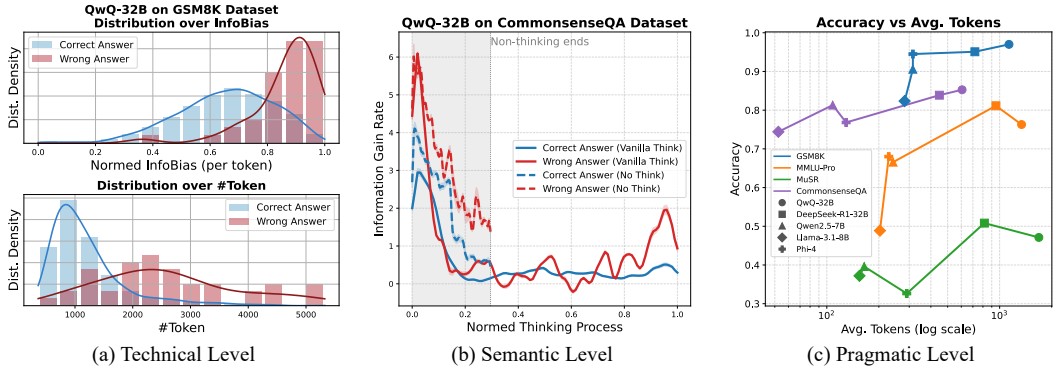

(a) Technical Level      (b) Semantic Level      (c) Pragmatic Level

Figure 1: **Understanding thinking inefficiency via Shannon & Weaver's Communication Model.** (a) Technical Level: On the GSM8K dataset, incorrect answers exhibit higher InfoBias and longer token lengths, suggesting that longer reasoning does not necessarily lead to better outcomes. (b) Semantic Level: The InfoGain rate shows a nonlinear decline as the thinking progresses, indicating diminishing contribution to entropy reduction over the final answer space. (c) Pragmatic Level: Results across various models and benchmarks show longer reasoning yields diminishing returns and may even reduce final accuracy. Detailed calculation methods and analysis are provided in §3.

enhance robustness against perturbations [Min et al., 2022]. However, once the reasoning length exceeds the model's reasoning capacity—an analogue to channel capacity—additional redundancy ceases to improve accuracy and instead induces error accumulation and semantic drift (Figure 1(a)). **At the semantic level**, as the number of reasoning steps increases, the information gain per step rapidly diminishes; excessive reasoning contributes little to uncertainty reduction and may even introduce semantic noise, revealing inefficiencies in the mapping between symbols and meanings (Figure 1(b)). **At the pragmatic level**, while longer reasoning chains may improve interpretability, they impose higher computational and cognitive costs, often yielding diminishing returns [Sprague et al., 2025] or even performance degradation on various tasks (Figure 1(c)).

This multi-level inefficiency highlights a central contradiction in the current LRM reasoning paradigm: substantial compute investments do not consistently translate into semantic efficiency or downstream performance gains. Motivated by this insight, we pose a core question: **Can we optimize the reasoning patterns of LRMs to substantially shorten reasoning chains while maintaining performance across diverse reasoning tasks?**

To quantitatively assess the efficiency of a model's reasoning process, we adopt an information-theoretic perspective and conduct in-depth analysis at two levels: (i) the response-level information bias, where we compute the mutual information between the model's generated response and the ground-truth reasoning trajectory to estimate InfoBias, capturing the overall semantic alignment across the full reasoning output (§3.2); (ii) the step-level information gain, where we quantify InfoGain as the entropy reduction over the answer distribution induced by each reasoning step, reflecting how much new information is introduced at each stage of the reasoning process (§3.3). Our empirical experiments (§3.4) reveal a significant, nonlinear positive correlation between reasoning length and InfoBias. Notably, incorrect answers consistently exhibit higher InfoBias, and the lengths of their generated responses are often biased toward longer reasoning chains. Furthermore, step-wise analysis indicates that models often possess a degree of intuitive confidence about the correct answer even before any explicit reasoning occurs. As reasoning unfolds, the InfoGain over the answer space and the model's confidence in the correct answer evolve differently across various types of reasoning tasks. While non-reasoning modes yield higher InfoGain per step, they typically result in lower overall confidence in the final answer compared to their reasoning-enabled counterparts.

Based on these analyses, we propose an **entropy-based Adaptive Think** strategy that dynamically halts reasoning once the model's confidence—quantified via entropy over the answer space—exceeds a tunable threshold (§4). We compare this approach against three alternative strategies: **Vanilla Think**, **No-Think**, and **Gated Think**. Extensive experiments (§5.2) are conducted across five language models and six benchmarks covering diverse types of reasoning tasks. Experimental results demonstrate that our Adaptive Think improves both accuracy and reasoning efficiency across

mathematical, factual, logical, and commonsense reasoning tasks. On two math benchmarks of varying difficulty, our method reduces token usage by 58.78% while preserving or slightly improving accuracy (average +0.95%). Beyond math, it boosts model accuracy by an average of 0.38% and reduces token usage by 42.39% across non-mathematical reasoning tasks. Finally, we conduct an in-depth analysis (§5.3) of when and how much reasoning a model should perform.

## 2 Related Work

**Information-Theoretic Perspectives** Information theory has long served as a lens for analyzing machine learning systems, particularly in studying generalization bounds [Russo and Zou, 2016, Xu and Raginsky, 2017] and understanding learning objectives [Slonim et al., 2002]. Recent work extends these ideas to LLMs, using entropy-based measures to evaluate reasoning reliability [Ton et al., 2024, Gan et al., 2025]. Semantic entropy, in particular, has been proposed as a tool for detecting hallucinations by measuring variability in meaning across generations [Farquhar et al., 2024], and can be efficiently estimated using hidden states alone [Kossen et al., 2024]. Other approaches use entropy to identify reasoning failures in multi-step generation without requiring supervision [Ali et al., 2025].

**Adaptive and Efficient Reasoning** Efficiency in LLMs is an active area of research, with methods that adapt step counts based on task difficulty, confidence, or resource constraints [Han et al., 2024, Pan et al., 2024, Shen et al., 2025]. Early exit mechanisms and dynamic token allocation [Yang et al., 2025a, Qu et al., 2025] aim to reduce unnecessary computation, while approaches such as elastic CoT and multi-scale reasoning seek to better align model capacity with problem complexity [Ma et al., 2025b, Kirchner et al., 2024]. Studies have shown that longer CoT do not always improve performance [Wu et al., 2025, Yang et al., 2025b, Yong et al., 2025], and in some cases can lead to overthinking, particularly in high-capacity models [Chen et al., 2024]. This has led to interest in minimal or even implicit reasoning strategies [Ma et al., 2025a, Sui et al., 2025], emphasizing the need for more nuanced reasoning strategies and adaptive control over reasoning depth.

## 3 Quantifying Thinking Efficiency

This section introduces a formal framework to measure reasoning efficiency by segmenting the thought process, analyzing divergence from ideal reasoning paths, and computing stepwise information gains.

### 3.1 Semantic Segmentation of Thinking Processes

Human reasoning typically unfolds in discrete, sequential steps [Guo et al., 2025b]. The means–ends analysis framework [Simon and Newell, 1971] views problem solving as a series of goal-subgoal transitions, each representing a cognitive operation. Similarly, ACT-R [Anderson et al., 1997, Whitehill, 2013] models reasoning as rule-based production sequences, while dual-process theory [Kahneman, 2011] characterizes "System 2" reasoning as deliberate and decomposable. Collectively, these theories motivate modeling reasoning as a structured sequence of semantically meaningful steps.

Accordingly, we segment a model's output reasoning path $S$ into discrete semantic units $S = \{s_1, s_2, \cdots, s_n\}$, where each $s_i$ represents a minimal step that contributes semantically to the overall process. For example, "solving $2x + 5 = 15$" triggers steps as $s_1$: subtract 5 from both sides $\rightarrow$ $s_2$: divide both sides by $2 \rightarrow s_3$: solve for $x$. These segments serve as the atomic elements for downstream information-theoretic analysis. The segmentation can be performed based on syntactic cues (e.g., clause or sentence boundaries), manual annotation, or automated approaches such as LLM-assisted chunking. By operating at this granularity, we enable a finer analysis of how incremental reasoning steps influence uncertainty and information flow throughout the trajectory.

### 3.2 Response-Level: Measuring Information Bias in Entire Trajectories

While $S$ captures the model's observable reasoning path, we posit the existence of a latent, ideal trajectory $T = \{t_1, t_2, \cdots, t_m\}$ representing the correct reasoning steps for a given question $Q$. This ideal trajectory may correspond to a human-annotated, cognitively plausible reasoning path, or reflect implicit reasoning steps within the model itself [Gan et al., 2025], which may differ from its explicit outputs. To measure how closely the model's reasoning aligns with this ground truth, we introduce

information bias, a metric based on mutual information:

$$\text{InfoBias}(S, T) = -I(s_{1:n}, t_{1:m}) = H(s_{1:n}, t_{1:m}) - H(s_{1:n}) - H(t_{1:m}), \tag{1}$$

where $I$ denotes mutual information and $H$ represents entropy. This discrepancy can be estimated via sampling, under the assumption that the generated reasoning trajectories $s$ and $t$ are two conditionally independent stochastic processes, and their joint distribution can be approximated through $N$ samples. Applying the KL-based estimation of mutual information [Paninski, 2003], we derive the following upper bound on the information bias:

$$|\hat{I}_N(S, T) - I(S, T)| \leq \sqrt{\frac{2 \log(2/\delta)}{N}} + \mathcal{O}\left(\frac{1}{N}\right), \tag{2}$$

where $\delta$ denotes the confidence level. This bound guarantees that the empirical estimate $\hat{I}_N(S, T)$ converges to the true mutual information $I(S, T)$ as $N$ increases, establishing InfoBias as a statistically consistent metric. Crucially, this enables reliable estimation of the alignment between observable and latent reasoning trajectories using a finite number of sampled inference steps.

### 3.3 Step-Level: Measuring Information Gain at Each Step

Beyond the trajectory as a whole, at the semantic level, we aim to quantify how each individual reasoning step contributes to answer inference. Efficient reasoning should progressively reduce uncertainty over the answer space [Sui et al., 2025]. Given a set of candidate answers $A = \{a_1, a_2, \cdots, a_l\}$, we can compute the conditional entropy at step $i$:

$$H_i = -\sum_{k=1}^{l} P(a_k|Q; s_{1:i}) \log P(a_k|Q; s_{1:i}), \tag{3}$$

where $P(a_k|Q; s_{1:i})$ is estimated from the model's output probabilities. Specifically, we concatenate the given question $Q$, the model's intermediate reasoning steps $s_{1:i}$, and the final answer prompt to form the input sequence (See §C.3 for details). The model's predicted probability of the next token is then used as the basis for evaluation. The information gain at step $i$ is:

$$\Delta I_i = H_{i-1} - H_i, \tag{4}$$

which quantifies how much uncertainty is reduced by incorporating step $s_i$. This reflects the extent to which each reasoning step clarifies the answer distribution. We further define a **targeted information gain** with respect to the correct answer $c \in A$:

$$\Delta I_i^c = -\log P(c|Q; s_{1:i}) - (-\log P(c|Q; s_{1:i-1})) = \log \frac{P(c|Q; s_{1:i-1})}{P(c|Q; s_{1:i})}, \tag{5}$$

capturing how each step influences the model's confidence in the correct option. Together, $\Delta I_i$ and $\Delta I_i^c$ reveal fine-grained reasoning efficiency, highlighting impactful steps toward the correct answer.

### 3.4 Empirical Evaluation and Insights

We empirically validate the methods proposed in §3.2 and §3.3, which respectively target the response-level relationship between reasoning length and InfoBias, and the step-level impact of individual reasoning steps on InfoGain. These analyses aim to assess the effectiveness of the information-theoretic metrics in capturing the dynamics and quality of reasoning exhibited by LLMs.

#### 3.4.1 InfoBias and the Risks of Overgeneration

To examine the relationship between response length and semantic deviation, we compute InfoBias over samples drawn from both model generations and reformulated ground-truth rationales (see §C.2 for details). Results on GSM8K (Figure 2) reveal two key observations:

**Findings 1: Cumulative InfoBias with Increased Reasoning Length.** We observe a consistent monotonic trend: longer reasoning chains tend to accumulate deviation from the correct reasoning path, suggesting that additional tokens often introduce noise rather than refinement. This pattern holds for both reasoning and non-reasoning models (see §D.1 for more results). There is no sign of

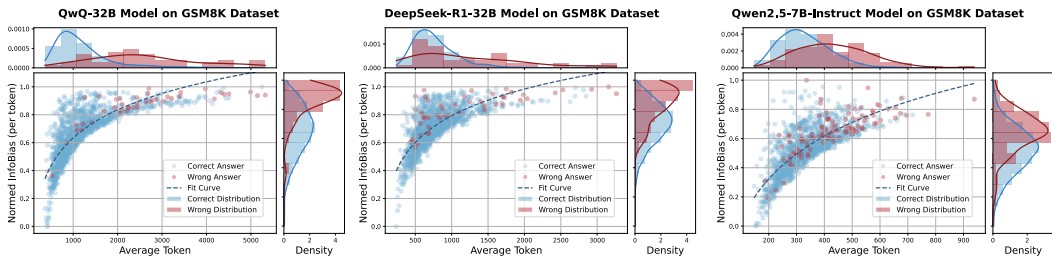

Figure 2: **Normalized InfoBias per token as a function of average reasoning length for different models on the GSM8K dataset.** Blue and red points represent instances with correct and incorrect answers, respectively, with density estimates of tokens and InfoBias shown on the top and right.

InfoBias saturation or decline even strong models exhibit rising bias, implying that simply generating more tokens does not guarantee improved alignment or correctness.

**Findings 2: Incorrect answers exhibit higher InfoBias and more variable response length.** A pronounced separation is observed between correct and incorrect samples: incorrect answers show higher InfoBias and slightly longer reasoning chains, indicating that extended reasoning amplifies rather than corrects misalignment. Moreover, the length distribution of incorrect answers is broader, indicating greater variability and instability in how models diverge from the correct reasoning path.

### 3.4.2  InfoGain and Step-Level Reasoning Quality

We next turn to the dynamics of reasoning steps. By segmenting rationales into paragraph-level units and measuring per-step InfoGain, we analyze how entropy and confidence evolve during inference across multiple benchmarks (Figure 3). Based on further analysis, we draw the following findings.

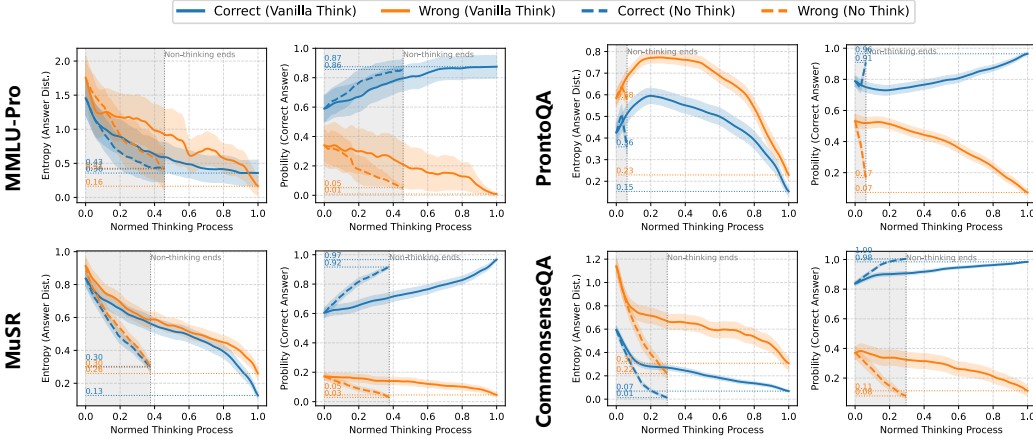

Figure 3: **Uncertainty dynamics across different reasoning benchmarks for QwQ-32B.** Each set includes two subplots: (1) entropy of the answer distribution vs. normalized reasoning steps, and (2) model-predicted probability of the correct answer over the same steps. Blue/orange lines denote correct/incorrect predictions; solid/dashed lines correspond to Vanilla Think and No-Think. Shaded areas mark the average token proportion used in No-Think mode. Step-wise analysis shows that models often exhibit early intuitive confidence in correct answers, even before reasoning starts. As reasoning unfolds, uncertainty decreases and confidence grows in task-specific ways.

**Findings 3: Reasoning Steps Consistently Reduce Uncertainty.** We observe that reasoning traces leading to correct answers consistently exhibit a reduction in entropy over the answer space and a corresponding increase in confidence for the correct choice. This supports the notion that effective reasoning incrementally filters uncertainty and sharpens prediction. Moreover, while No-Think mode achieves higher information efficiency per step—rapidly lowering entropy—it typically converges to lower final confidence, limiting its reliability. By comparison, Vanilla Think involves longer and less

efficient reasoning chains in terms of information gain per step, but ultimately yields more confident and accurate predictions, underscoring a trade-off between efficiency and robustness in reasoning.

**Findings 4: Reasoning Models Exhibit Initial Intuition.** Even before reasoning begins (step = 0), samples that eventually lead to correct answers already show lower entropy and higher confidence. This indicates that the models possess an initial bias or "intuitive prior" toward the correct answer even before engaging in multi-step reasoning. This effect is especially pronounced in knowledge-intensive tasks like MMLU-Pro and CommonsenseQA, suggesting that LRMs often start with strong inductive biases toward the correct choice, possibly due to extensive prior exposure during training.

**Findings 5: Task-Specific Reasoning Dynamics.** In CommonsenseQA, entropy drops rapidly at the early stages, suggesting that commonsense questions can often be resolved with minimal reasoning. Notably, No-Think mode yields higher final confidence than Vanilla Think, implying that the latter's intermediate reasoning steps may be redundant or inefficient. Meanwhile, MMLU-Pro and MuSR show smooth and monotonic entropy separation between correct and incorrect samples, reflecting tasks where gradual semantic integration is beneficial. In contrast, ProntoQA exhibits a non-monotonic pattern—entropy first rises, then falls—which may result from its binary format: early steps broaden the hypothesis space and reduce overconfidence before eventual convergence. Overall, these dynamics reflect how the task's type influence the utility of the reasoning process.

These findings highlight the potential of entropy-based signals as proxies for monitoring and controlling reasoning in LRMs. The steady accumulation of InfoBias with longer reasoning suggests that unregulated generation often leads to semantic drift, while InfoGain trends reveal diminishing returns from extended reasoning. Early confidence signals also suggest that further reasoning is often unnecessary. These insights motivate our approach: adaptively modulating reasoning depth based on entropy, allowing models to think when needed and stop when additional steps offer little value.

# 4   Entropy-Based Adaptive Thinking

Modern LRMs differ fundamentally from earlier non-reasoning models in both training and inference paradigms. Traditional models were typically trained with task-specific supervision to imitate step-by-step reasoning implicitly [Trung et al., 2024, Pang et al., 2025], while modern LRMs are increasingly trained via reinforcement learning to develop general-purpose reasoning capabilities [Guo et al., 2025a]. At inference time, these models no longer rely solely on internal heuristics but instead generate explicit reasoning traces, often marked by structured tokens such as `<think>` and `</think>`. This shift enables more controllable and interpretable reasoning, opening new avenues for modulating the reasoning process dynamically. Based on this paradigm, we design and evaluate several distinct reasoning modes, as shown in Figure 4.

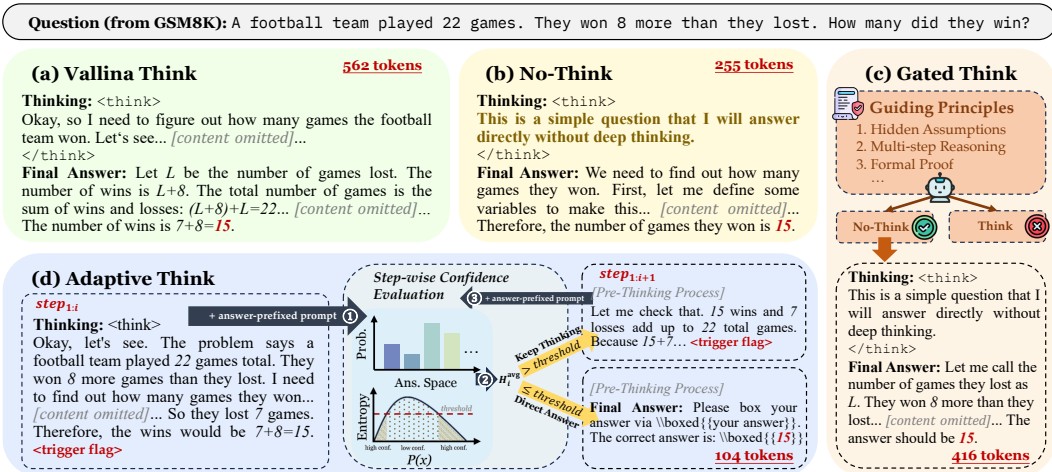

Figure 4: **An illustration of four thinking modes on a sample question from the GSM8K dataset.**

**(a) Vanilla Think.** It represents the model's default reasoning pattern, in which it first engages in an extended chain-of-thought process in response to a given question, generating intermediate reasoning steps before eventually producing a final answer based on the full context of its prior thinking.

**(b) No-Think.** While most current reasoning models are designed to perform detailed reasoning before producing an answer, it is possible to steer the model toward bypassing this process by modifying the chat template. A common strategy involves forcing the thinking box to remain empty during decoding [Team, 2025a]. However, we find that using the following prompt more effectively encourages the model to adopt a non-reasoning mode when generating its response.

```
<think>
This is a simple question that I will answer directly without deep thinking.
</think>
```

**(c) Gated Think.** This setting represents a hybrid of the Vanilla Think and No-Think modes. Given a question, the model is prompted to first assess whether deep thinking is necessary—typically performing this assessment in a no-think mode. To guide this process, we design a heuristic framework that considers several factors, such as whether the question requires inference beyond surface-level cues, involves multi-step reasoning or information synthesis, demands rigorous logical or mathematical justification, presents multiple plausible strategies, or calls for hypothesis-driven analysis. Based on this assessment, the model proceeds in either deep thinking or direct-answer mode. Detailed criteria and prompt are provided in the §C.4.

**(d) Adaptive Think.** Empirical results in §3.4 reveal that information bias with respect to the correct reasoning path tends to accumulate as the response length increases. Each reasoning step contributes to reducing entropy over the answer space and increasing confidence in the correct answer, forming clear trends. Since entropy reflects the model's uncertainty over the answer distribution, we propose an Adaptive Think strategy to dynamically decide when to terminate reasoning. After each intermediate reasoning step, the model computes the average entropy $H_i^{\mathrm{avg}} = \frac{1}{l} \sum_{i=1}^{l} H_i$ over the answer space. Reasoning is terminated early once the average entropy falls below a confidence threshold, which is parameterized by a hyperparameter $\alpha \in [0, 1]$—with smaller values of $\alpha$ corresponding to stricter entropy thresholds. To formalize this, we note that the entropy of a discrete distribution (i.e., the function $-p \log_2 p$) is upper-bounded by $1/e \ln 2$ when $p \in (0, 1]$. Using this bound, we define the following stopping criterion at the $i$-th step:

$$\begin{cases} \text{Output the final answer directly} & \text{if } H_i^{\mathrm{avg}} \leq \alpha \cdot \frac{1}{e \ln 2} \\ \text{Continue reasoning} & \text{otherwise} \end{cases}. \tag{6}$$

When the model is determined to have reached sufficient confidence and no further thinking is needed, we follow the approach introduced by Muennighoff et al. [2025], prompting the model to generate a final response by appending an `</think>` tag—used only when the model is still within the thinking phase—followed by an answer-prefixed prompt to elicit the final output.

## 5 Experiments

### 5.1 Experimental Settings

**Models and Datasets.** We conduct comprehensive experiments using 8 language models—three reasoning-augmented models (QwQ-32B and DeepSeek-R1-Distill-Qwen-7B/32B) and five standard models (LLaMA3.1-8B-Instruct, Phi-4, Qwen2.5-7B/32B-Instruct, and Yi-1.5-34B-Chat)—across six diverse benchmarks, including two mathematical datasets of varying difficulty and four benchmarks covering distinct types of reasoning tasks, including GSM8K, AIME2025, MMLU-Pro, MuSR, ProntoQA, and CommonsenseQA. Details are provided in the §C.1.

**Implementation Details.** We employ the high-throughput inference engine vLLM [Kwon et al., 2023] to support efficient model reasoning. For all methods, each question is evaluated with five independent inference runs, and the results are averaged to ensure robustness. For all datasets, we either adopt the standard prompts provided in their original papers or construct task-specific prompts tailored to our setting. Further implementation details, including dataset-specific prompt templates and answer extraction methods, are described in the §C.5.

## 5.2 Main Results

We conduct experiments on two math reasoning benchmarks with different difficulty levels: GSM8K (standard) and AIME2025 (challenging), as shown in Table 1. First, reasoning models outperform non-reasoning ones significantly. For instance, on AIME2025, QwQ-32B with Vanilla Think achieves 70.67% accuracy, far surpassing Phi-4's 20.00%. However, this comes at the cost of much higher token usage—reasoning models consume on average 3.4× more tokens on GSM8K and 9.5× more on AIME2025 compared to non-reasoning baselines. Additionally, the more challenging AIME2025 benchmark results in substantially higher average token consumption than GSM8K.

Table 1: **Performance and efficiency comparison on two math reasoning benchmarks.** Models are evaluated based on accuracy (where higher is preferred) and average token count (where lower is preferred) across four different strategies: Vanilla Think, No-Think, Gated Think, and Adaptive Think. The comparison encompasses both non-reasoning and reasoning models, offering a thorough analysis of the trade-offs between reasoning performance and computational efficiency.

| Models | Think Mode | GSM8K | | AIME2025 | |
| --- | --- | --- | --- | --- | --- |
| | | Acc ↑ | #Token ↓ | Acc ↑ | #Token ↓ |
| *Non-Reasoning Models* | | | | | |
| Llama-3.1-8B-Instruct | Base | 82.35 | 281.90 | 0.00 | 1015.37 |
| | CoT | 81.83 | 295.95 | 0.00 | 1201.35 |
| Qwen2.5-7B-Instruct | Base | 90.58 | 314.62 | 8.00 | 802.10 |
| | CoT | 90.55 | 318.55 | 5.33 | 854.14 |
| Phi-4 | Base | 94.50 | 314.96 | 13.33 | 1388.26 |
| | CoT | 94.77 | 335.11 | 20.00 | 1536.40 |
| Yi-1.5-34B-Chat | Base | 82.73 | 310.93 | 0.00 | 646.04 |
| | CoT | 82.18 | 313.95 | 0.00 | 919.05 |
| Qwen2.5-32B-Instruct | Base | 95.24 | 292.68 | 10.00 | 797.88 |
| | CoT | 94.63 | 304.26 | 10.67 | 830.06 |
| *Reasoning Models* | | | | | |
| DeepSeek-R1-Distill-Qwen-7B | Vanilla Think | 85.47 | 433.08 | 45.33 | 10885.24 |
| | No-Think | 87.85 | 264.79 | 8.67 | 957.08 |
| | Gated Think | 87.34 | 279.76 | 26.67 | 5958.27 |
| | Adaptive Think | **87.95** | **259.80** | **47.33** | **4188.40** |
| | Δ *vs. Vanilla* | *+2.90%* | *-40.01%* | *+4.41%* | *-61.52%* |
| DeepSeek-R1-Distill-Qwen-32B | Vanilla Think | 95.09 | 718.81 | 56.67 | 9527.68 |
| | No-Think | 93.18 | 253.48 | 22.67 | 2321.91 |
| | Gated Think | 94.24 | 296.75 | 53.33 | 7705.33 |
| | Adaptive Think | **95.98** | **356.30** | **57.33** | **4765.15** |
| | Δ *vs. Vanilla* | *+0.94%* | *-50.43%* | *+1.16%* | *-49.99%* |
| QwQ-32B | Vanilla Think | 97.00 | 1132.32 | 70.67 | 14595.73 |
| | No-Think | 97.00 | 758.22 | 68.00 | 13290.79 |
| | Gated Think | 96.81 | 824.55 | 70.00 | 13941.63 |
| | Adaptive Think | **97.73** | **379.80** | **71.33** | **4633.50** |
| | Δ *vs. Vanilla* | *+0.75%* | *-66.46%* | *+0.93%* | *-68.25%* |

Next, we compare three thinking modes: Vanilla Think, No-Think, and Gated Think. Skipping the reasoning step (No-Think) drastically reduces token usage but also degrades accuracy, especially for DeepSeek-R1-32B. Gated Think offers a trade-off between accuracy and efficiency, falling between Vanilla and No-Think in both metrics, indicating modest gains in efficiency at the cost of performance. Finally, our proposed Adaptive Think strategy outperforms all thinking modes. It slightly improves accuracy across both benchmarks while reducing token usage by an average of 56.11%, effectively pruning redundant reasoning and significantly enhancing efficiency.

In addition, Table 2 presents results on four additional benchmarks spanning diverse reasoning types. On all tasks, the entropy-based Adaptive Think consistently outperforms Vanilla Think for QwQ-32B, with an average accuracy gain of 1.23% and a 42.52% reduction in token usage. For DeepSeek-R1-32B, Adaptive Think reduces average token consumption by 49.43%. We observe a slight drop in accuracy on the MMLU-Pro and MuSR datasets compared to Vanilla Think. We hypothesize that this may be due to DeepSeek-R1-32B being a distilled model rather than one trained with reinforcement learning, which could limit its capacity for autonomous reasoning.

Furthermore, on CommonsenseQA, a benchmark focused on shallow, intuition-based reasoning, Adaptive Think achieves substantial improvements in efficiency. Specifically, QwQ-32B with

Table 2: **Performance and efficiency comparison on four other reasoning benchmarks.**

| Models | Think Mode | MMLU-Pro | | MuSR | | ProntoQA | | CommonsenseQA | |
|---|---|---|---|---|---|---|---|---|---|
| | | Acc ↑ | #Token ↓ | Acc ↑ | #Token ↓ | Acc ↑ | #Token ↓ | Acc ↑ | #Token ↓ |
| *Non-Reasoning Models* | | | | | | | | | |
| Llama-3.1-8B-Instruct | Base | 48.86 | 202.96 | 37.20 | 154.55 | 85.32 | 333.49 | 74.41 | 52.44 |
| | CoT | 50.57 | 429.68 | 39.26 | 322.62 | 90.72 | 434.58 | 73.82 | 216.46 |
| Qwen2.5-7B-Instruct | Base | 66.57 | 241.80 | 39.60 | 164.54 | 97.84 | 339.09 | 81.31 | 108.17 |
| | CoT | 66.00 | 433.11 | 39.97 | 330.68 | 98.68 | 385.27 | 80.59 | 233.23 |
| Phi-4 | Base | 68.00 | 228.29 | 32.62 | 290.15 | 99.44 | 271.44 | 76.82 | 129.33 |
| | CoT | 72.86 | 872.34 | 33.73 | 633.97 | 99.60 | 337.97 | 78.78 | 283.13 |
| Yi-1.5-34B-Chat | Base | 51.43 | 173.11 | 40.26 | 112.16 | 87.20 | 363.33 | 71.20 | 84.74 |
| | CoT | 54.29 | 444.46 | 42.78 | 358.38 | 91.07 | 433.95 | 68.31 | 241.79 |
| Qwen2.5-32B-Instruct | Base | 75.71 | 188.69 | 44.23 | 144.97 | 99.87 | 222.85 | 86.17 | 76.63 |
| | CoT | 80.86 | 375.51 | 46.51 | 302.17 | 99.73 | 276.56 | 85.36 | 203.74 |
| *Reasoning Models* | | | | | | | | | |
| DeepSeek-R1-Distill-Qwen-7B | Vanilla Think | 57.43 | 1287.57 | 39.79 | 1036.89 | 93.60 | 1082.80 | 62.95 | 481.40 |
| | No-Think | 40.29 | 223.87 | 37.96 | 97.29 | 76.80 | 372.69 | 53.66 | 62.98 |
| | Gated Think | 41.43 | 234.59 | 38.62 | 172.63 | 72.00 | 406.34 | 52.19 | 89.36 |
| | Adaptive Think | | | | | | | | |
| | - $\alpha = 0.1$ | **58.57** | **722.84** | **41.40** | **725.35** | **93.67** | **979.80** | **63.02** | **266.98** |
| | $\Delta$ *vs. Vanilla* | *+1.99%* | *-43.86%* | *+4.05%* | *-30.05%* | *+0.07%* | *-9.51%* | *+0.11%* | *-44.54%* |
| DeepSeek-R1-Distill-Qwen-32B | Vanilla Think | 81.14 | 951.68 | 50.82 | 815.76 | 98.76 | 621.69 | 83.87 | 447.86 |
| | No-Think | 68.00 | 208.91 | 44.68 | 152.66 | 96.20 | 241.80 | 80.95 | 102.40 |
| | Gated Think | 77.14 | 245.41 | 46.30 | 306.20 | 98.40 | 433.42 | 80.92 | 105.20 |
| | Adaptive Think | | | | | | | | |
| | - $\alpha = 0.1$ | **79.43** | **521.33** | **50.03** | **568.26** | **99.64** | **543.79** | 84.54 | 136.30 |
| | - $\alpha = 0.2$ | 78.57 | 414.25 | 48.84 | 428.13 | 97.28 | 447.66 | **84.60** | **85.16** |
| | - $\alpha = 0.3$ | 74.57 | 336.22 | 47.25 | 325.77 | 94.72 | 351.30 | 84.28 | 54.73 |
| | $\Delta$ *vs. Vanilla* | *-2.11%* | *-45.22%* | *-1.55%* | *-30.34%* | *+0.89%* | *-12.53%* | *+0.87%* | *-80.99%* |
| QwQ-32B | Vanilla Think | 76.29 | 1338.95 | 47.12 | 1685.59 | 99.36 | 1167.05 | 85.27 | 606.19 |
| | No-Think | 76.29 | 612.17 | 42.38 | 634.91 | 98.76 | 697.75 | 85.00 | 179.00 |
| | Gated Think | 78.57 | 674.57 | 44.31 | 643.70 | 98.80 | 890.42 | 85.09 | 177.43 |
| | Adaptive Think | | | | | | | | |
| | - $\alpha = 0.1$ | **77.14** | **629.33** | **47.86** | **1077.76** | **99.96** | **882.64** | **86.68** | **262.90** |
| | - $\alpha = 0.2$ | 76.86 | 443.69 | 46.11 | 729.11 | 96.68 | 677.54 | 86.52 | 159.81 |
| | - $\alpha = 0.3$ | 74.00 | 317.34 | 44.87 | 500.66 | 94.72 | 532.08 | 85.81 | 88.51 |
| | $\Delta$ *vs. Vanilla* | *+1.11%* | *-53.00%* | *+1.57%* | *-36.06%* | *+0.60%* | *-24.37%* | *+1.65%* | *-56.63%* |

$\alpha = 0.1$ reduces token usage by 56.63%, while DeepSeek-R1-32B achieves an even larger reduction of 80.99%, both compared to their respective Vanilla Think baselines. These tasks typically rely on basic elimination strategies and commonsense priors rather than complex, multi-step reasoning, which often leads to redundant or exploratory computational paths. Adaptive Think's entropy-based control halts early once confidence suffices, reducing costs with minimal accuracy impact.

## 5.3 In-Depth Analysis

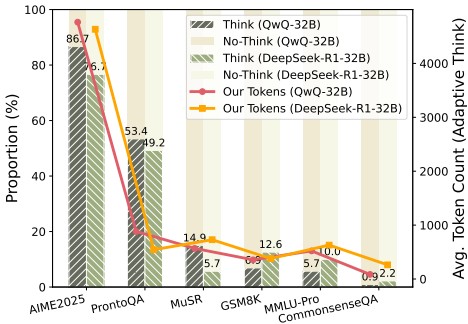
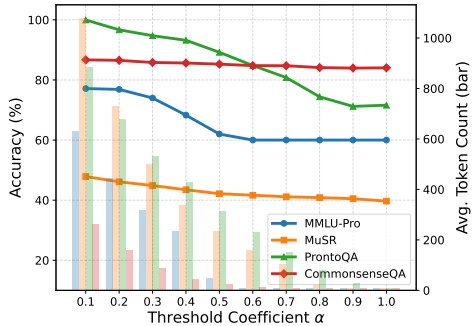

Figure 5: Proportion of think vs. no-think samples in Gate Think mode and corresponding token usage under Adaptive Think.

Figure 6: Effect of parameter $\alpha$ on accuracy and token count, showing the trade-off between reasoning performance and efficiency.

**To Think or Not to Think?** We analyze the "think" vs. "no-think" decisions under the *Gate Think* setting to assess the model's ability to adapt reasoning to task difficulty in Figure 5. On AIME2025, which requires strong mathematical reasoning, QwQ-32B and DeepSeek-R1-32B engage in "think" mode for 86.7% and 76.7% of samples, respectively. In contrast, for CommonsenseQA dominated

by superficial commonsense cues, these proportions drop to 0.9% and 2.2%. This demonstrates the models' ability to selectively allocate reasoning based on task complexity. Notably, the average token count under Adaptive Think mirrors this pattern, with more computation allocated to harder tasks. This reflects the core strength of *Adaptive Think*: it dynamically adjusts reasoning effort to match problem difficulty, improving efficiency without compromising performance.

**How Much Thinking is Enough?**  We further examine how varying the confidence threshold coefficient $\alpha$ in *Adaptive Think* impacts accuracy and token efficiency across tasks (Figure 6). Results show that optimal reasoning depth is task-dependent. For logic- and knowledge-intensive benchmarks such as ProntoQA and MMLU-Pro, higher thresholds are critical—premature stopping leads to significant accuracy drops (e.g., from 99.96% to 71.60% on ProntoQA and from 77.14% to 60.00% on MMLU-Pro). These tasks demand deeper reasoning to resolve ambiguity or recall fine-grained knowledge. In contrast, soft-reasoning tasks such as CommonsenseQA and MuSR exhibit greater robustness to early stopping. Due to their reliance on surface-level cues or redundant contextual information, these tasks allow models to make confident decisions early in the reasoning process. As a result, increasing $\alpha$ leads to minimal accuracy degradation while significantly reducing token consumption, highlighting opportunities for efficiency gains in low-complexity scenarios.

## 6   Limitations and Future Work

**Model and Task Selection Constraints**  Adaptive Think requires access to a model's next-token probability distribution, so we evaluate it on open-source deployments that expose this interface. For closed-source models, such as OpenAI's o1, we can only employ the sampling-based approximation method from Farquhar et al. [2024] to estimate the answer-space distribution, limiting us to analytical assessments of reasoning efficiency. While prior methods focus on multiple-choice tasks, we broaden Adaptive Think to free-response benchmarks like GSM8K and AIME2025. Leveraging a tree-search algorithm to derive answer-space distributions allows us to rigorously measure entropy reduction through model reasoning. However, for truly open-ended questions where no single "correct" answer exists, Adaptive Think cannot yet optimize reasoning efficiency, and, to our knowledge, no existing work has tackled this challenge. We outline this as a key avenue for future investigation.

**From Output- to Model-Oriented Optimization**  Adaptive Think offers a flexible, plug-and-play mechanism that dynamically reduces unnecessary reasoning steps and sequence length during inference. This stands in sharp, fundamental contrast to model-based efficient reasoning, which seeks to compress full-length reasoning models into more concise variants or to train inherently efficient reasoning architectures from scratch. While Adaptive Think provides a lightweight approach to mitigate a model's tendency toward excessive overthinking, model-based methods directly enhance the core reasoning capacity and efficiency of the model itself. However, reducing inference cost solely through output manipulation leaves largely unaddressed the underlying architectural and algorithmic inefficiencies that limit scalability and adaptability across tasks. Transitioning from output-oriented to model-oriented optimization is therefore crucial: by redesigning model internals, such as attention mechanisms, intermediate representation formats, and gradient flow pathways, we can achieve more substantial, generalizable, and sustainable gains in reasoning speed, resource usage, and performance consistency. Accordingly, our next phase of work will investigate model-centric techniques for deeper and more robust improvements in inference efficiency.

## 7   Conclusion

This paper revisits inefficient reasoning in LRMs through an information-theoretic lens. While extended reasoning chains are often used to improve accuracy, we find that longer outputs often lead to higher bias and semantic redundancy. By introducing InfoBias and InfoGain, we reveal that excessive reasoning often introduces semantic redundancy with limited benefit. Building on these insights, we introduce an entropy-based Adaptive Think strategy that dynamically halts reasoning once confidence is sufficiently high, enabling models to allocate effort based on task complexity while maintaining competitive accuracy. Experiments across diverse tasks and models show that Adaptive Think offers a promising trade-off between efficiency and performance, allowing models to reason selectively—thinking more when necessary, and less when intuition suffices.

# 8 Acknowledgements

This work was supported by the Public Computing Cloud at Renmin University of China and the Fund for Building World-Class Universities (Disciplines) at Renmin University of China.

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

# Appendix

# A  Related Work

**Information-Theoretic Analyses of Language Models**   Information theory provides a principled foundation for analyzing machine learning systems, especially in understanding generalization, uncertainty, and learning dynamics. Classical works apply mutual information and related measures to characterize generalization performance in deep learning [Russo and Zou, 2016, Xu and Raginsky, 2017], as well as to clarify the structure of unsupervised learning objectives [Slonim et al., 2002] and summarization tasks [West et al., 2019].

More recently, these tools have been adapted to LLMs, where entropy-based methods help characterize and diagnose model behavior. For instance, Ton et al. [2024] and Gan et al. [2025] use information-theoretic frameworks to study the alignment and reliability of LLM reasoning. A notable development is the introduction of semantic entropy, which quantifies the variability of meaning across possible generations to detect hallucinations and semantic inconsistencies [Farquhar et al., 2024]. Semantic Entropy Probes (SEPs) further improve efficiency by estimating this uncertainty directly from intermediate hidden states, without requiring multiple generations [Kossen et al., 2024]. Beyond output quality, entropy has also been used to investigate the internal reasoning process of LLMs. Ali et al. [2025], for example, propose entropy-based probes to localize failure points in multi-step reasoning chains, offering a supervision-free alternative to error attribution.

**Efficiency and Adaptivity in LLM Reasoning**   As reasoning tasks become more complex and LLMs more capable, managing the cost-performance trade-off has emerged as a critical research theme. Several works propose adaptive reasoning frameworks that dynamically adjust the number of reasoning steps (e.g., chain-of-thought length) based on input difficulty or intermediate confidence [Han et al., 2024, Pan et al., 2024, Shen et al., 2025, Xu et al., 2024]. These approaches often aim to reduce unnecessary computation while preserving answer quality. Complementary to this, early exit strategies enable models to halt generation once sufficient evidence or confidence has been gathered [Yang et al., 2025a, Damani et al., 2024, Wang et al., 2025]. Other resource-aware designs include complexity-aware token allocation[Qu et al., 2025] and routing across model sizes or reasoning depths[Kirchner et al., 2024], aligning inference cost with task demands.

A parallel line of research revisits the assumption that longer reasoning is always better. Empirical studies show diminishing or even negative returns from overly long CoTs, particularly in high-capacity models prone to spurious logic and hallucinations [Wu et al., 2025, Yang et al., 2025b, Chen et al., 2024]. This "overthinking" phenomenon suggests that the optimal reasoning depth is both task- and model-dependent [Su et al., 2025]. Interestingly, short or minimal CoTs often achieve comparable performance, especially outside symbolic or mathematical domains [Ballon et al., 2025, Jin et al., 2024, Sprague et al., 2025].

Lastly, some recent perspectives challenge the necessity of explicit chain-of-thought reasoning altogether. Studies show that models can exhibit reasoning-like behavior even without intermediate thought supervision or explicit stepwise prompts [Ma et al., 2025a, Sui et al., 2025]. These findings reinforce the need for flexible, confidence-aware mechanisms that can dynamically modulate reasoning depth—potentially without committing to rigid CoT formats.

# B  Theoretical Proofs

## B.1  Proof of Equation (2)

This section provides a complete derivation of the upper bound for the deviation between the empirical mutual information estimator and the true mutual information, as discussed in Equation 2. This derivation is based on the analysis of mutual information estimation presented by Paninski [2003].

Consider the empirical mutual information estimator $\hat{I}_N(S, T)$ computed from $N$ i.i.d. samples $(s_i, t_i)_{i=1}^N$ drawn from the joint distribution $p(s, t)$. The estimator is defined as

$$\hat{I}_N(S, T) = \hat{H}_N(S) + \hat{H}_N(T) - \hat{H}_N(S, T), \tag{7}$$

where $\hat{H}_N(\cdot)$ are empirical entropy estimates (e.g., the kernel-based entropy estimator used in our experiments). The deviation $\left|\hat{I}_N(S, T) - I(S, T)\right|$ arises from two sources:

- **Bias.** The estimator $\hat{I}_N(S, T)$ can be biased for finite $N$, especially when $N$ is small.
- **Variance.** Random fluctuations occur due to finite-sample variability.

**Concentration of measure.** The variance component can be controlled using McDiarmid's inequality, which generalizes Hoeffding's inequality to functions of independent random variables with bounded differences. For any $\epsilon > 0$,

$$P\left(\left|\hat{I}_N - \mathbb{E}[\hat{I}_N]\right| \geq \epsilon\right) \leq 2 \exp\left(-\frac{2N\epsilon^2}{c^2}\right), \tag{8}$$

where $c$ is a constant representing the sum of squared bounds on the differences. In our setting, $c = \mathcal{O}(1)$.

Setting the confidence level to $1 - \delta$ and solving for $\epsilon$:

$$\epsilon \propto \sqrt{\frac{2\log(2/\delta)}{N}}. \tag{9}$$

**Bias of the estimator.** Paninski [2003] provides a bound on the bias of the entropy estimator, showing that

$$\mathbb{E}[\hat{I}_N(S, T)] - I(S, T) = \mathcal{O}\left(\frac{1}{N}\right), \tag{10}$$

which accounts for the bias term in the deviation.

**Total deviation.** Combining the variance bound from Equation 9 with the bias bound in Equation 10, we obtain the overall deviation between the empirical and true mutual information with probability at least $1 - \delta$:

$$\left|\hat{I}_N(S, T) - I(S, T)\right| \leq \sqrt{\frac{2\log(2/\delta)}{N}} + \mathcal{O}\left(\frac{1}{N}\right). \tag{11}$$

This provides the theoretical justification for the upper bound used in Equation 2 of the main text, which captures both the stochastic variability and the systematic bias of the mutual information estimator.

## B.2 Computational Overhead of Adaptive Think

This section provides a rigorous theoretical analysis of the computational overhead introduced by the proposed **Adaptive Think** strategy, compared to the standard **Vanilla Think** decoding.

**Notation.** We use the following notation:

- $C_m$: compute cost per generated token (dominated by large matrix multiplications on GPU).
- $T$: average number of generated tokens per reasoning step.
- $S_v$: average number of reasoning steps under Vanilla Think.
- $S_a$: average number of reasoning steps under Adaptive Think (early stopping), $S_a \leq S_v$.
- $L_v = S_v T$: total number of tokens under Vanilla Think.
- $L_a = S_a T$: total number of tokens under Adaptive Think.
- $r_t = 1 - \frac{L_a}{L_v} = 1 - \frac{S_a}{S_v}$: token reduction rate.
- $C_e$: cost of extracting probabilities and computing entropy at each reasoning step (lightweight scalar/vector operations).
- $N$: tree search depth (free-form generation only).
- $W$: beam width (free-form generation only).

We assume scalar/vector operations are typically $10^3$–$10^4 \times$ faster than matrix multiplications on GPU, hence

$$\frac{C_e}{C_m} \approx 10^{-3}. \tag{12}$$

**Vanilla Think.** The total computational cost of Vanilla Think is the cost of generating all tokens:

$$C_{\text{vanilla}} = C_m L_v = C_m S_v T. \tag{13}$$

**Multiple-choice tasks.** For multiple-choice tasks, Adaptive Think performs entropy-based early stopping. The total cost consists of:

- Generation cost: $C_m S_a T$.
- Entropy estimation overhead: $S_a C_e$.

The total cost is

$$C_{\text{MC}} = C_m S_a T + S_a C_e. \tag{14}$$

For Adaptive Think to be more efficient than Vanilla Think, we require

$$
\begin{aligned}
C_{\text{MC}} < C_{\text{vanilla}} &\iff C_m S_a T + S_a C_e < C_m S_v T \\
&\iff \frac{S_a}{S_v}\left(1 + \frac{C_e}{C_m T}\right) < 1 \\
&\iff r_t > \frac{\frac{C_e}{C_m T}}{1 + \frac{C_e}{C_m T}}.
\end{aligned}
\tag{15}
$$

Using Equation 12, suppose $T = 20$. Then $\frac{C_e}{C_m T} = 5 \times 10^{-5}$, and the required $r_t$ is approximately $0.005\%$. In practice, any nonzero token reduction already leads to net savings. This shows that entropy estimation overhead is negligible compared to matrix operations, and computational savings are approximately equal to the token reduction rate $r_t$.

**Free-form generation tasks.** For free-form generation, Adaptive Think may perform a tree search to determine early stopping. At each step, this introduces additional matrix computations equivalent to generating $W \times N$ extra tokens. The total cost becomes

$$C_{\text{FF}} \approx C_m S_a T + S_a (C_m W N) = C_m S_a (T + W N). \tag{16}$$

Requiring $C_{\text{FF}} < C_{\text{vanilla}}$ gives

$$
\begin{aligned}
C_m S_a (T + W N) < C_m S_v T &\iff \frac{S_a}{S_v} < \frac{T}{T + W N} \\
&\iff r_t > \frac{W N}{T + W N}.
\end{aligned}
\tag{17}
$$

This provides a clear **critical threshold** for $r_t$: the token reduction rate must exceed $\frac{W N}{T + W N}$ for Adaptive Think to yield computational savings.

**Numerical example.** In our experiments, the minimum reasoning length was $T = 120$, tree depth $N = 10$, and beam width $W = 5$. Then

$$\frac{T}{T + W N} = \frac{120}{120 + 50} = 0.7059 \quad \Rightarrow \quad r_t > 1 - 0.7059 = 0.2941 \ (29.41\%).$$

The observed average token reduction was $58.78\%$, well above this threshold, confirming that Adaptive Think leads to significant latency reductions even with tree search overhead.

**Practical considerations.**

- **No KV-cache reconstruction or mode switching.** Entropy estimation and tree search are performed within the standard decoding loop, avoiding any system-level latency from switching between prefill and decoding modes.
- **Lightweight entropy estimation.** Scalar operations are negligible compared to matrix multiplications, making the overhead minimal for multiple-choice tasks.
- **Controlling $W$ and $N$.** Equation 17 gives clear design guidance: increasing $T$ or reducing $W$ and $N$ lowers the required $r_t$ threshold for net savings.

**Summary.**

- For **multiple-choice tasks**, the entropy estimation overhead is negligible, and any token reduction ($r_t > 0$) leads to overall savings.

- For **free-form generation**, tree search introduces extra matrix computation proportional to $WN$. Computational savings occur when

$$r_t > \frac{WN}{T + WN}.$$

- In practice, Adaptive Think achieves substantial efficiency improvements without requiring modifications to the standard decoding pipeline.

## C  Implementation Details

### C.1  Models and Datasets

We conduct comprehensive experiments using three reasoning-augmented models:

- **QwQ-32B** [Team, 2025b]: a 32B-parameter language model developed by Alibaba's Qwen team, emphasizing advanced reasoning capabilities. It features a 32K token context length and demonstrates performance comparable to OpenAI's o1 model on several benchmarks. The model is designed to embody principles of curiosity and reflection, aiming to enhance analytical reasoning during responses.

- **DeepSeek-R1-Distill-Qwen-7B/32B** [Guo et al., 2025a]: a distilled version of the DeepSeek-R1 model, fine-tuned on synthetic data generated by the original R1 model. This variant leverages the Qwen architecture and benefits from reinforcement learning techniques to enhance reasoning capabilities. The distillation process aims to retain the reasoning strengths of DeepSeek-R1 while improving efficiency and accessibility.

To evaluate the generality and robustness of our methods, we employ six reasoning-focused benchmarks spanning diverse domains and cognitive requirements. Below, we provide detailed descriptions of each dataset:

- **GSM8K** [Cobbe et al., 2021] (Elementary mathematics)
  **Reasoning Type:** Multi-step numerical reasoning
  **Description:** GSM8K is a dataset of 8.5K high-quality, linguistically diverse grade school math word problems. Solving each question typically requires several steps of numerical reasoning, often involving intermediate arithmetic operations. The dataset has become a standard testbed for evaluating the chain-of-thought capabilities of language models, especially in structured, algorithmic domains.

- **AIME2025** (Advanced competition mathematics)
  **Reasoning Type:** Symbolic, multi-step, and abstract reasoning
  **Description:** This benchmark consists of problems from the 2025 American Invitational Mathematics Examination (AIME) I and II, a prestigious U.S. math competition for high school students. Compared to GSM8K, AIME2025 features significantly higher problem complexity, demanding more abstract algebraic manipulation, geometric insight, and symbolic reasoning. It serves as a rigorous test of deep mathematical reasoning.

- **MMLU-Pro** [Wang et al., 2024] (General knowledge and academic reasoning)
  **Reasoning Type:** Knowledge-intensive multi-hop reasoning
  **Description:** MMLU-Pro is an enhanced version of the original MMLU benchmark [Hendrycks et al., 2020], which contains questions from 57 diverse academic subjects. MMLU-Pro focuses on more complex, multi-hop questions that test both factual knowledge and the ability to integrate information across domains. It is designed to reflect real-world professional exam scenarios, such as medical, legal, or scientific reasoning.

- **MuSR** [Sprague et al., 2024] (Narrative comprehension)
  **Reasoning Type:** Soft, multi-step reasoning over long contexts
  **Description:** MuSR (Multi-step Soft Reasoning) features long-form ( 1,000 words) natural

language narratives, requiring the model to reason about evolving relationships, causality, and world knowledge. Each instance poses a series of questions that depend on the entire story, emphasizing temporal coherence, contextual memory, and soft inference rather than purely symbolic logic.

- **ProntoQA** [Saparov and He, 2023] (Logical deduction)
  **Reasoning Type:** Symbolic and deductive reasoning
  **Description:** ProntoQA is a synthetic benchmark constructed to systematically evaluate deductive reasoning capabilities in LLMs. The dataset includes logic puzzles framed in natural language, each of which has a unique, deterministically derivable correct answer. It is especially useful for probing consistency, error propagation, and how well models can follow logical implications.

- **CommonsenseQA** [Talmor et al., 2018] (Commonsense knowledge)
  **Reasoning Type:** Heuristic and intuitive reasoning
  **Description:** CommonsenseQA challenges models to answer questions that require everyday commonsense understanding, typically in the absence of direct textual evidence. It evaluates a model's ability to use prior knowledge and intuitive judgment to select the most plausible answer among distractors, making it a key benchmark for human-aligned reasoning.

## C.2 Information Bias Calculation Settings

First, we utilize the GSM8K dataset, prompting each model to generate 10 responses per question to obtain a sample-based estimation of the random variable $S$. To ensure consistency in response generation, we employ the following standardized reasoning prompt across all evaluated LLMs:

```
Please answer the question step by step. Remember to box your final answer via $\\
    boxed{your answer}$. If there is no correct answer, give a random answer.
```

Furthermore, to obtain correct reasoning paths $T$ for each question, we leverage the fact that GSM8K provides gold-standard step-by-step solutions. Following the approach in Gan et al. [2025], we use the Llama3.1-70B-Instruct model to paraphrase each gold solution 10 times. The resulting paraphrases are aggregated to form a sample-based estimation of the random variable $T$. For paraphrasing the ground-truth answers, the following prompt was used:

```
You will be given a problem-solving process. Please rewrite this process without
    changing its logic or content. Ensure that the output includes only the
    rewritten process and nothing else.

Problem-Solving Process: {input}

Rewritten Process:
```

Subsequently, we estimate the mutual dependence between the random variables $S$ and $T$ using the Hilbert-Schmidt Independence Criterion (HSIC). HSIC provides a non-parametric measure of statistical dependence by projecting the data into a reproducing kernel Hilbert space and quantifying the cross-covariance between the transformed variables. We employ the Gaussian kernel for this purpose, as it offers greater expressiveness for capturing complex, nonlinear relationships compared to linear or inverse multiquadratic (IMQ) kernels. To determine an appropriate bandwidth parameter $\sigma$, we follow a common heuristic based on the median of pairwise Euclidean distances among samples.

To mitigate potential bias introduced by varying response lengths, we normalize the raw HSIC scores by the number of tokens in each response. This normalization yields a per-token dependency measure, allowing for equitable comparison. The overall setup is specifically designed to evaluate the alignment between model-generated reasoning paths and reference derivations.

## C.3 Information Gain Calculation Settings

To accurately estimate the conditional probability distribution over the final answer space at a specific stage of reasoning—as well as the corresponding entropy—we concatenate the question $Q$, the intermediate reasoning steps up to step $i$, denoted as $s_{1:i}$, and an answer-prompting suffix. This composite prompt is then used to compute the token-level probabilities of subsequent model outputs.

For instance, in the case of multiple-choice questions, we illustrate this procedure using an example from the MMLU-Pro dataset:

```
Question: The symmetric group $S_n$ has $\n\\factorial{n}$ elements, hence it is not
    true that $S_{10}$ has 10 elements.\nFind the characteristic of the ring 2Z.\
    nA. 0\nB. 30\nC. 3\nD. 10\nE. 12\nF. 50\nG. 2\nH. 100\nI. 20\nJ. 5

<think>
Okay, so I need to find the characteristic of the ring 2Z and choose the correct
    option from the given list. Hmm, let me recall what the characteristic of a
    ring means. The characteristic is the smallest positive integer n such that n
    times the multiplicative identity (1) equals zero in the ring. But wait, 2Z is
    the ring of even integers, right? So first, does 2Z have a multiplicative
    identity? ...[content omitted]...
</think>

The correct option letter is:
```

For open-ended question answering tasks such as GSM8K and AIME2025, we adopt a different approach. Given a constructed answer-prompt suffix, we perform a tree-based search over the space of possible continuations. Starting from the prompt, we recursively expand the output tokens until a termination symbol (e.g., a closing brace "}") is generated, marking a complete candidate answer path. The probability of each candidate answer is computed as the product of the conditional probabilities of the tokens along its corresponding path. Finally, all candidate sequences are ranked by their total probabilities, and the top-$K$ most probable completions are selected as final answer candidates.

```
Question: Steve and Tim decide to see who can get home from school the fastest.
    Steve lives further away than Tim, so he is allowed to ride his bike. Steve
    lives 3 miles from the school and can bike at 440 feet per minute. Tim lives 2
    miles away from the school. If Tim can ride his skateboard at 264 feet per
    minute, how long will the winner be waiting at their house before the loser
    finishes the race?

<think>
Okay, let's see here. So the problem is about Steve and Tim racing to their homes
    from school. I need to figure out how long the winner will have to wait for the
     loser. Hmm, right. Let me parse the details again. First, Steve lives farther
    away and he bikes at 440 feet per minute. Tim lives closer, only 2 miles away,
    but skates his skateboard at 264 feet per minute. The question is, who gets
    home first, and then how much time the winner has to wait? Alright, so I think
    the key here is to calculate the time each person takes to get home and then
    find the difference between those times. The person with the shorter time is
    the winner, and the difference will be how long they have to wait. ...[content
    omitted]...
</think>

Please box your final answer via \\boxed{{your answer}}. The correct answer is: \\
    boxed{{
```

For the visualization method described in §3.4.2, we first normalize the number of reasoning steps for each question in the Vanilla Think mode to a 0–1 range and plot the line chart of metric changes across all questions. We then fit separate curves for correctly and incorrectly answered questions, along with their 95% confidence intervals (shaded regions represent the confidence bands). For the No Think mode, we compute the average number of output tokens per benchmark and scale the curve proportionally based on its ratio to the average reasoning length in Vanilla Think mode, ensuring a consistent comparison of the overall reasoning process.

### C.4 Detailed Criteria for Gated Think

In cognitive and educational psychology, a substantial body of work suggests that the structural features of a question—rather than its superficial difficulty—are key predictors of whether deep human reasoning is likely to be invoked. Specifically, when a problem cannot be answered through direct

retrieval or one-step logic, humans tend to engage in multi-hop reasoning, hypothesis generation, and information synthesis, as exemplified by contrastive policy learning approaches in symbolic domains [Poesia et al., 2021]. Such cognitively demanding tasks activate deeper, sequential processes involving intermediate inference steps. In formal domains like mathematics and logic, tasks that require proofs—such as induction, contradiction, or recursion—are known to elicit structured chains of reasoning [Besold et al., 2021]. When a problem presents multiple valid solution paths or requires the evaluation of competing strategies, humans instinctively perform mental simulations of plans, assess outcomes, and prune suboptimal branches—hallmarks of strategic reasoning [Hao et al., 2023]. Lastly, tasks that involve multi-variable relationships, such as scientific hypothesis testing or economic modeling, often require systematic modeling, assumption tracking, and iterative validation. These processes map closely to "System 2" reasoning and have recently been formalized in comprehensive causal reasoning benchmarks—CausalBench, for instance, evaluates LLMs' ability to identify and reason about cause-and-effect structures across diverse domains [Zhou et al., 2024].

Taken together, these findings suggest that the need for deep reasoning is not determined solely by surface difficulty but by structural complexity—such as requirements for synthesis, recursion, proof, and planning. Based on this understanding, we introduce the following prompt to operationalize this judgment process. The five criteria outlined are derived from well-established cognitive demands associated with deeper human reasoning.

---

**Decision Criteria for Triggering Deep Think Mode**

You are an intelligent reasoning assistant. Upon receiving a question, you must determine whether it requires **Deep Think Mode**—which involves rigorous, multi-step, and systematic complex reasoning.

**Evaluation Criteria (At least TWO must be met to trigger Deep Think Mode):**

1. **Cannot be answered directly based on the question itself**

   - The answer is not immediately apparent from general knowledge, simple reasoning, or single-step calculations.
   - The question requires combining multiple knowledge points, hidden conditions, or assumptions.

2. **Multi-step reasoning & information integration**

   - The solution involves sequential logical steps, where each step depends on previous conclusions.
   - Multiple data sources, conditions, or assumptions must be synthesized to derive the final answer.

3. **Strict mathematical/logical proof or recursive deduction**

   - The problem requires formal proof (e.g., deductive reasoning, axiomatic proofs).
   - It involves recursive reasoning, mathematical induction, or constructing counterexamples.

4. **Non-trivial strategy or non-unique solution**

   - The question requires evaluating multiple potential solutions and **choosing the optimal one**.
   - There may be multiple valid approaches, requiring deep analysis and comparison.

5. **Systematic reasoning & hypothesis-based deduction**

   - The question requires establishing hypotheses and systematically deriving conclusions.
   - Multiple variables and complex relationships are involved, requiring a rigorous analytical process.

**Output Format:**

   - **"YES" (Deep Think Mode required)** If the question meets at least 2 criteria, return "YES" and briefly explain why.

- **"NO" (Deep Think Mode not required)** If the question only requires basic or short-step reasoning, return "NO" and explain why it can be answered directly.

**Examples**
**Requires Deep Think**

- **Input**: *"Let A, B, and C be three sets. Prove that $A \cap (B \cup C) = (A \cap B) \cup (A \cap C)$."* **Output**: "YES – This problem involves set operations and requires a formal mathematical proof with multi-step logical deductions."

- **Input**: *"If the speed of light is the cosmic limit, but the universe is expanding, is it possible for two regions to be permanently unobservable from each other?"* **Output**: "YES – This question involves relativity, cosmology, and hypothesis-based deduction, requiring systematic reasoning."

- **Input**: *"On an 8x8 chessboard, if two opposite corners are removed, can it be completely covered by 2x1 dominoes?"* **Output**: "YES – This requires constructing a counterexample, analyzing the board's parity, and recursive reasoning."

**Does Not Require Deep Think**

- **Input**: *"What is $2^{10}$?"* **Output**: "NO – This is a straightforward computation that can be answered directly."

- **Input**: *"Tom is 5 years older than Alice. Alice is 10 years old. How old is Tom?"* **Output**: "NO – This is a basic arithmetic problem that does not require complex reasoning."

- **Input**: *"Why is water heavier than oil?"* **Output**: "NO – This is a factual question about density that can be answered using common knowledge."

## C.5 Experimental Setup and Implementation Details

We employ the high-throughput inference engine vLLM as the execution framework, with generation hyperparameters set to a temperature of 0.8, top-p of 1.0, and a repetition penalty of 1.05. For all evaluated methods, we conduct five independent inference runs per question. The final accuracy and token usage are computed by averaging across these runs.

For our Adaptive Think approach, we segment the model's reasoning trajectory into discrete steps based on paragraph boundaries. Specifically, the occurrence of a double newline ("\n\n") is treated as a trigger flag, prompting an entropy-based decision on whether to continue the reasoning process. To mitigate noise and ensure meaningful intermediate content, we enforce a minimum length of 120 characters per reasoning step.

It is worth noting that for the two mathematical benchmarks—GSM8K and AIME2025—we define the answer candidate space using a $top - K = 5$. Candidate answer sequences are retrieved via a tree search procedure (detailed in §C.3), where token continuations are explored recursively. The maximum tree depth is capped at 10, which is sufficient to accommodate the full range of answer lengths found in both benchmarks.

# D Additional Experimental Results

## D.1 InfoBias vs. Reasoning Length

Figure 7 broadens our evaluation by incorporating two additional models—Llama 3.1-8B-Instruct and Phi-4—and uncovers the same systematic pattern identified in Figure 2. Specifically, as the length of the reasoning chain increases, the generated outputs progressively diverge from the ground-truth solution, highlighting a clear trade-off: each extra token tends to introduce cumulative noise. Quantitatively, we observe a steady rise in solution InfoBias metrics with longer inference trajectories. Crucially, this information drift is not confined to specialized reasoning architectures but also plagues general-purpose models, underscoring the pervasive challenge of semantic drift in current

large-language systems. These results motivate the need for both output-level interventions—like adaptive chain-length control—and deeper, model-centric optimizations to mitigate drift at its source.

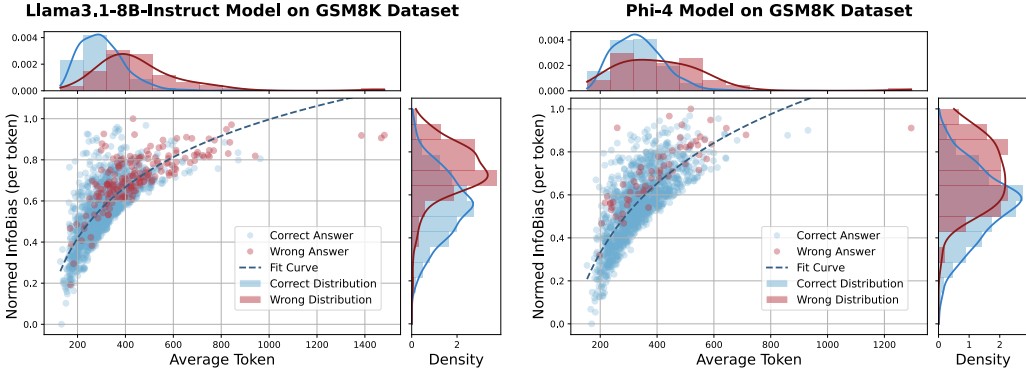

Figure 7: **Normalized InfoBias per token as a function of average reasoning length for Llama3.1-8B-Instruct and Phi-4 on the GSM8K dataset.** Blue and red points represent instances with correct and incorrect answers, respectively, with density estimates of tokens and InfoBias shown on the top and right. Each subplot illustrates the relationship between reasoning length and InfoBias.

## D.2 InfoGain per Reasoning Step

Under the same experimental setup as QwQ-32B, Figure 8 plots how uncertainty-related metrics evolve throughout the reasoning process of eepSeek-R1-Distill-32B across four distinct types of reasoning tasks.

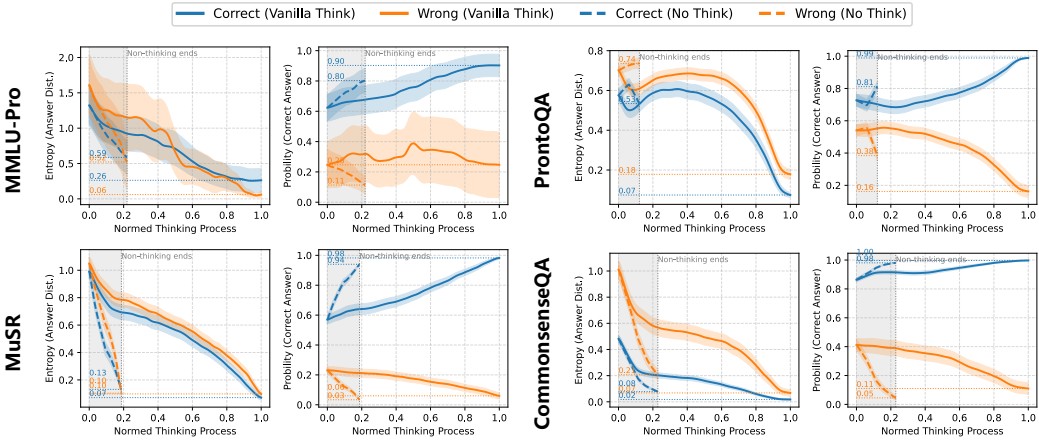

Figure 8: **Uncertainty dynamics across different reasoning benchmarks for DeepSeek-R1-Distill-32B.** It presents similar thinking dynamics on various benchmarks with QwQ-32B.

## D.3 Analysis of Performance Across Question Difficulty Levels

Table 3 shows the performance of QwQ-32B on the MATH500 under different levels of difficulty.

Table 3: **Results of QwQ-32B on the MATH500 Dataset.**

| Think Mode | Total | | Level-1 | | Level-2 | | Level-3 | | Level-4 | | Level-5 | |
|---|---|---|---|---|---|---|---|---|---|---|---|---|
| | Acc | #Token | Acc | #Token | Acc | #Token | Acc | #Token | Acc | #Token | Acc | #Token |
| **Vanilla Think** | 69.80 | 3893.76 | 81.40 | 1444.51 | 76.67 | 2039.04 | 71.43 | 2921.08 | 69.53 | 4058.89 | 60.45 | 6529.85 |
| **Adaptive Think** | 70.80 | 1506.31 | 83.72 | 648.56 | 78.89 | 793.60 | 73.33 | 1087.34 | 68.75 | 1412.85 | 61.19 | 2677.82 |
| *Δ vs. Vanilla* | *+1.43%* | *-61.31%* | *+2.85%* | *-55.10%* | *+2.90%* | *-61.08%* | *+2.66%* | *-62.78%* | *-1.12%* | *-65.19%* | *+1.22%* | *-59.00%* |

Experimental results demonstrate that Adaptive Think substantially reduces reasoning length across most difficulty levels, achieving an average token reduction of 61.31%, while also enhancing overall accuracy with an average improvement of 1.43%. The performance gains are especially pronounced at low to medium difficulty levels, where accuracy improves by more than 2.5% on Levels 1 to 3. Although there is a slight decrease in accuracy at Level 4, amounting to 1.12%, the method still achieves more than 65% reduction in reasoning tokens, indicating that Adaptive Think effectively optimizes reasoning efficiency even for more complex problems.

Overall, these findings further validate the adaptability and efficiency of our approach across varying levels of question difficulty.

# E  Case Studies

In this section, we showcase model reasoning traces from diverse datasets, including case studies where Vanilla Think produced incorrect answers but Adaptive Think, through efficient reasoning optimization, arrives at the correct answers. In the following Vanilla Think reasoning traces, we highlight redundant or irrelevant content—such as repeated or unnecessarily expanded information, background details unrelated to the question, and inconsequential discourse markers—in gray. Critical reasoning steps that lead to incorrect final answers are highlighted in red to indicate misleading or erroneous directions. Since the model already achieves high accuracy on GSM8K and ProntoQA, we selected case studies from the other benchmarks. Each chosen example was answered incorrectly in all five trials under the Vanilla Think mode but correctly under the Adaptive Think mode.

## E.1  A case from MMLU-Pro

---

**Case 1: A wrong response from MMLU-Pro based on the Vanilla Think**

**Question:** The study of reality in the broadest sense, an inquiry into the elemental nature of the universe and the things in it, is known as __?
A. quantum physics
B. metaphysics
C. ontology
D. phenomenology
E. cosmology
F. teleology
G. epistemology
H. axiology
I. N/A

**Model Answer:**
`<think>`
Okay, let's tackle this question. The user is asking about the study of reality in the broadest sense, specifically an inquiry into the elemental nature of the universe and its contents. The options given are from A to J, with some being philosophy branches and others specific sciences.
First, I need to recall what each term means. Let me start with the options one by one.
Quantum physics (A) is a branch of physics dealing with particles at a microscopic level, so probably not the answer here since it's more specific. Metaphysics (B) comes to mind; I remember that metaphysics deals with the fundamental nature of being and the world. Ontology (C) is a part of philosophy that studies being, existence, and reality. So that's very close. Phenomenology (D) is about structures of experience and consciousness, so maybe not. Cosmology (E) is the study of the universe's origin and structure, which is more specific again. Teleology (F) refers to purposes or causes, like the study of design or purpose in nature. Epistemology (G) is about knowledge and justified belief, so that's different. Axiology (H) pertains to values, like ethics and aesthetics. The rest are N/A.
The question specifies "the elemental nature of the universe and the things in it." That sounds like ontology, which focuses on being and existence. But sometimes metaphysics is considered the broader category that includes ontology. Wait, is ontology a subset of metaphysics? Maybe

the answer could be either B or C. Let me think again. The question says "the study of reality in the broadest sense" and "inquiry into the elemental nature". Metaphysics as a whole covers topics like being, existence, time, causality, etc., while ontology is specifically about being and reality's basic elements. However, sometimes people might refer to metaphysics as the broader field. Hmm.

Wait, the exact terms: Metaphysics is often defined as the branch of philosophy investigating the first principles of things, including cosmology, ontology, and teleology. Ontology specifically deals with 'what exists', so if the question is about the elemental nature, ontology might be the direct answer. However, the phrasing "study of reality in the broadest sense" might align more with metaphysics.

Alternatively, let me check the standard definitions. According to philosophy sources, ontology is the study of being and existence, while metaphysics also includes other areas. Since the question mentions "elemental nature of the universe," maybe ontology is the precise term here. But sometimes metaphysics is used more broadly. Let me see examples. For instance, when philosophers talk about metaphysical questions, they do include things like what is real, the fundamental constituents of the universe, etc. So maybe metaphysics (B) is the right answer because it encompasses ontology within it.

Alternatively, if the question is pointing directly at the study of existence, then ontology. But I think the common answer expected here would be ontology (C)... So maybe the answer is ontology (C).

Wait, perhaps the difference is subtle... Therefore, if the question is specifically about the nature of reality's elements, ontology is more precise. But maybe the question considers metaphysics as the broader term.

Let me think of how these terms are taught. In Western philosophy, metaphysics is often called the study of the fundamental nature of reality, being and the world. So perhaps the answer is metaphysics (B). Wait now I'm confused between B and C.

Let me parse the question again: "the study of reality in the broadest sense, an inquiry into the elemental nature of the universe and the things in it" — so that sounds exactly like the definition of metaphysics. For example, when people say metaphysics deals with the fundamental nature of reality, existence, space, time, etc. Ontology is a major branch of metaphysics. So the broader term is metaphysics. Therefore, the correct answer should be B. Hmm, but maybe the question is tricky. Let me think again.

Wait, the question says "the elemental nature of the universe and the things in it" — elemental in terms of what constitutes them, their basic building blocks. Is that ontology or metaphysics? If it's the elements (like atoms, or substances), that's part of ontology. But metaphysics overall includes those questions. Maybe the best way is to go with the standard textbook answer.

For example, according to standard definitions, metaphysics is the branch concerned with the fundamental nature of reality. Ontology is a part of metaphysics that deals with existence. So the answer is B, metaphysics. But I might have mixed this up. Let me try to recall a similar question.

In the GRE or LSAT, such questions often name metaphysics as the study of the fundamental nature of reality. Yes, I think that's the case. So the correct answer is B. But wait, the options have both B and C. Let me confirm once more. Suppose the question is...

Alternatively, when someone asks about the fundamental nature of the universe and everything in it, that's typically metaphysics. Since ontology is a subset...

Maybe the answer expected here is ontology (C). Because ontology is the study of existence and reality's basic elements. Alternatively, metaphysics (B) is the broader term.

Let me look for another angle. The term "ontology" is derived from the Greek "onton" (beings) and "logia" (study). So ontology is the study of beings/existence. Metaphysics is a broader field that includes ontology, cosmology, etc. So if the question is specifically about the study of the elemental nature (the basic elements that compose things), then that's ontology. Hence, the correct answer should be C. So I think I made a mistake earlier thinking it was B. So the answer is C. But I need to be certain.

Let me think again. If the question says... "study of reality in the broadest sense" — metaphysics is broader because it includes not just existence (ontology) but also other issues like causation, possibility. So maybe the answer is B. However, the second part specifies "inquiry into the elemental nature of the universe and the things in it" — that points to the

basic constituents and their nature, so ontology. Thus, perhaps the answer is C. I'm a bit torn, but I'll go with ontology (C).
`</think>`
After considering the definitions of each term and analyzing the question's phrasing, the correct answer is ontology (C). The question specifically refers to the "elemental nature of the universe and the things in it," which directly aligns with ontology's focus on the study of being, existence, and the fundamental categories of reality. While metaphysics (B) is a broader field that includes ontology, the question emphasizes the elemental aspects, making ontology the precise fit.

$\boxed{C}$

**Correct Answer: B**

---

### Case 1: A correct response from MMLU-Pro based on the Adaptive Think

**Question:** Same as above.

**Model Answer:**
`<think>`
Okay, so I need to figure out which term refers to the study of reality in the broadest sense, focusing on the elemental nature of the universe and its contents.
First, I remember that philosophy has several branches. Metaphysics is one of them. I think metaphysics deals with the fundamental nature of being and the world. Ontology is a part of metaphysics, right? Ontology is specifically about what exists and the nature of existence. So maybe the question is pointing towards metaphysics as a broader term?
Then there's epistemology, which is about knowledge. Phenomenology is the study of structures of consciousness or experience. Cosmology is more about the origin and structure of the universe, often in a scientific context. Teleology deals with purposes or causes. Axiology is about values. Quantum physics is a science, not philosophy.
The question says "the study of reality in the broadest sense" and "elemental nature of the universe and things in it." That sounds like metaphysics because metaphysics asks questions like "What is existence?" and "What is the fundamental structure of reality?" Ontology is part of that but more specific. Since the question mentions "broadest," maybe B (metaphysics) is better than C (ontology).
`</think>`
The correct option letter is: $\boxed{B}$

**Correct Answer: B**

---

### E.2 A case from MuSR

### Case 2: A wrong response from MuSR based on the Vanilla Think

**Question:** In a shocking twist of events, Charlotte is found lifeless in her kitchen, felled by a stun gun, and it's up to seasoned Detective Winston to unravel the mystery, with prime suspects being Letti and Deborah.
Detective Winston sat at a small, cluttered desk, scanning over the police reports and witness statements regarding the murder scene. His eyes narrowed in on a peculiar piece of information; Charlotte, the victim, had reportedly been stuffed.
"That's culinary," Winston muttered.
The morning began with a visit to the local pet store, where Letti, a groupie with a penchant for exotic decor, was often spotted buying various animals.
"You've seen this woman before?" he asked, flashing his badge and a photograph of Letti towards the shopkeeper.
"Oh, yeah! She comes by often," the shopkeeper said cheerfully. "Always looking for some new pet to tag along with her to those concerts, I guess."

"And what happens to these pets?"

The shopkeeper's face contorted, his smile fading slightly. "Well... They just disappear, don't they? It's odd."

His next stop was the scene of the murder: a secluded kitchen in Charlotte's home. According to her housekeeper, it was such place which Charlotte and Letti had been seen alone, sharing the early dinner Charlotte had invited her over for.

Suddenly, a lead brought him to a Pawn Shop known for its wide array of self-defense equipment, stun guns in particular. Winston slid the security footage into the player, Letti appearing on the screen, her face a match to the shopkeeper's description.

She was seen making a purchase, a stun gun catching his trained eye. Making an additional discovery, Winston found a newspaper article, along with some concert footage showing Letti using a stun gun effectively for self-defense during a music event. This was all coming together.

Next, he turned towards a report stating Letti had recently signed up for self-defense classes - the kind that teaches stun gun usage.

His trail returning him to the pet store, Winston had found out that Letti had been at Charlotte's earlier in the day, according to a tip from a disgruntled neighbor.

Upon hearing about the style in which the stuffing was done, detective Winston noted, Letti seemed to recognize it immediately. Could have been a coincidence, could have not; only time would tell.

Doctor Winston had his work cut out for him, a tangle of clues, and facts left behind, waiting to be pieced together. He could see, however, that the threads of this mystery were starting to knit together. The mystery deepened the more he dug into it, each fact, each clue pulling him further into the complex web spun around the suspect. The only thing left to do was to pull the right string and observe as everything fell into place. With every step, the truth was seemingly closer, yet shrouded with enough obscurity to keep him on his toes.

Winston was met with a distressed and nervous-looking woman when he arrived at Charlotte's house. Deborah was shaking, a pitiful sight, as she repeated her claim: "I was scared, I was really scared of her." She was a mess, with trembling hands and a trembling voice, but Winston had seen the proof of her fear in the way Charlotte treated her. Harsh treatment by a woman, once her employer and now her victim.

Apparently, Deborah had been scheduled to drive Charlotte that very day; a service she always offered. In all honesty, it seemed Deborah was Charlotte's only method of transportation. Winston couldn't help but wonder, though, at the intensity of her fear. Was it because of Charlotte's abusive behavior, evidenced by bruises and the testimonies of her neighbors? Or was it due to ominous messages that Charlotte had sent Deborah?

Deborah's fear seemed to have had a history of manifesting physically. The word was out that she had employed a stun gun in the past, for self-protection as Winston understood. Deborah was never far from it, the weapon always at her side.

Just then, Winston noticed a license on her wall. "Used to be a personal security guard, I see," he commented, pointing towards it. Deborah just nodded in agreement, a simple affirmative to Winston's observation. Strange how life can take unexpected turns, Winston thought.

"Deborah," Winston started, his tone soothing, "can you tell me what exactly happened when Charlotte... died?" Deborah hesitated, her eyes moving towards the stun gun on the table. An eerily similar device was collected from the crime scene.

"Well, I..." she stammered, "I, um, was the last person to see her. I... I found her... in the kitchen." Her voice was barely audible, a whisper barely able to carry the weight of her confession.

Winston didn't say anything. He simply nodded before standing up, thanking Deborah for her time and promising to be in contact. He left the house, lost in his thoughts. There was much to ponder over in this investigation. After all, the puzzle of the case was only just beginning to reveal itself.

Who is the most likely murderer?
A. 'Letti'
B. 'Deborah"

**Model Answer:**

<think>
Okay, let's try to figure this out. So the question is asking who the most likely murderer is between Letti and Deborah based on the given story. I'll need to go through all the details provided and see which suspect has stronger evidence pointing towards them.

First, starting with Letti. The key points here are:

- Letti is a frequent buyer at the pet store, getting exotic animals that "disappear." The mention of Charlotte being "stuffed" might be a clue here. Since Letti deals with animals that disappear, maybe she used a method related to that in the murder. But "stuffed" could also refer to the way Charlotte was found, perhaps implying she was literally stuffed like a doll? Not sure, but the detective joked it was "culinary," which might suggest something about the method.

- Letti bought a stun gun from the pawn shop, which matches the murder weapon. There's security footage showing her purchasing it. Also, there's concert footage of her using a stun gun effectively for self-defense. Plus, she enrolled in self-defense classes teaching stun guns. So she has access to the weapon and knows how to use it.

- The neighbor's tip says she was at Charlotte's earlier that day. Since the murder happened at Charlotte's house and the housekeeper mentioned they were seen alone for an early dinner, that places her there. So opportunity is there too.

Now, Deborah's points:

- She was Charlotte's driver and the only one providing transport. She was the last person to see Charlotte alive and found her in the kitchen. However, Deborah was very nervous and claimed she was scared of Charlotte. Testimonies mention Charlotte's abusive behavior towards her, leading to bruises and messages. So possible motive could be fear or retaliation?

- Deborah had a stun gun for self-protection, which was at the scene. The same type was found there. She's a former security guard, so maybe she knows how to use such weapons. Wait, the doctor found the stun gun at the scene, but the story mentions both Letti bought one and Deborah carries one. The weapon used was a stun gun, so both have access. But Letti's possession is documented through the pawn shop purchase, while Deborah's is part of her self-defense.

So Debora's alibi? She says she was the last to see Charlotte and found her. But if she was there, she could have done it. However, Deborah's motive might stem from being abused, but does that escalate to murder? Maybe. Letti's motive isn't clear yet. Why would Letti kill Charlotte? Perhaps related to the "stuffing"? Maybe they were involved in something illegal with animals, but Charlotte got in the way? Alternatively, maybe Letti wanted to steal something or there was a personal conflict?

Other clues: Letti was seen at Charlotte's for a dinner invitation. So they were together alone. That's a perfect opportunity. Deborah was supposed to drive Charlotte that day. If she drove her there, maybe after the dinner, but Deborah claims to have found Charlotte. Hmm.

Wait, the housekeeper saw Charlotte and Letti having an early dinner together. So the murder likely happened then. So Letti was there when Charlotte died. Deborah might have come later, or maybe she was present but not mentioned. Wait, the housekeeper said Charlotte and Letti were alone for the dinner. So Deborah might not have been there at the time of death. Unless Deborah had access as the driver. Maybe Deborah was the one who let Letti in or was present, but not sure.

Deborah's presence at the scene (found her) might mean she arrived afterward. But the murder weapon is a stun gun. Both have access. Letti purchased it, so maybe she used that. The murder weapon could be traced back to the pawn shop purchase. Deborah's stun gun is her own, but how would that connect?

The doctor noted Letti recognized the stuffing method, which might link her to knowing how Charlotte was positioned or something. But the story says "stuffing" was culinary, so maybe Letti's expertise in handling animals (like stuffing them?) could be relevant. So Charlotte being "stuffed" could be a play on words, implying Letti did something with preserving or preparing a body, but that's unclear.

Alternatively, the "stuffing" refers to how Charlotte was positioned, and Letti's familiarity with that suggests she did it. Meanwhile, Deborah's connection is that she found the body but might have been the one who killed her.

So putting it together: Letti has motive (maybe financial, or conflict?), opportunity (was there at dinner), means (bought the stun gun). Deborah's motive is abuse, but opportunity might

be less direct unless she was nearby. However, since the housekeeper saw Charlotte and Letti alone during dinner, that's a strong indication that Letti was present when Charlotte was killed.

Deborah's role as the driver and the last to see her might mean she arrived after the murder and found Charlotte, but if Deborah had the stun gun, maybe she came afterward and... Wait, but if Letti used the stun gun during the dinner, then let's see timing. Letti and Charlotte are alone. Letti uses the stun gun, kills Charlotte, then leaves. Then Deborah arrives later to drive her, finds her dead. But Deborah's part in that would be coincidental. Unless Deborah was part of it, but no indication yet.

Alternatively, Deborah could have acted out of fear, using her stun gun to attack Charlotte, perhaps when she was driving her, but the scene was Charlotte's kitchen. So Deborah might have gone there, arguing, using her stun gun. But why would she be there after the dinner? Maybe she crashed the dinner? The housekeeper only mentions Charlotte and Letti being there.

Another angle: Deborah's possession of a stun gun and being a security guard gives her knowledge. But Letti's recent purchase and training show intent. The concert footage shows Letti using one effectively, which supports her capability.

The fact that Deborah was the last to see Charlotte might imply she found the body, but maybe she did it and lied about it. However, Letti has stronger clues: the purchase, presence, and training. Therefore, the answer might be Letti (option A).
</think>

$\boxed{A}$

**Correct Answer: B**

---

**Case 2: A correct response from MuSR based on the Adaptive Think**

**Question:** Same as above.

**Model Answer:**
<think>
Okay, let's break this down. So Charlotte was found dead in her kitchen, killed by a stun gun. The main suspects are Letti and Deborah.

First, looking at Letti's clues: She's been buying a lot of animals from the pet store, but they disappear. The shopkeeper mentioned that. Then Winston saw security footage of her buying a stun gun at the pawn shop. Also, there's a newspaper article showing she used a stun gun before at a concert for self-defense. Plus, she signed up for self-defense classes that teach stun gun use. Oh, and the way Charlotte was "stuffed" might be a clue. The detective thought it was culinary, maybe implying Letti's familiarity with preparing animals (since she buys pets that disappear) led her to stuff Charlotte? That seems a bit odd though. Maybe the "stuffing" refers to how the stun gun was used, like in a specific manner that Letti knows? Now moving on to Deborah. She was Charlotte's driver and had a history of being abused by Charlotte. Witnesses say Charlotte treated her harshly, even leaving bruises. Deborah has a stun gun for self-defense, which she carries with her. She was the last person to see Charlotte alive and found her body. The murder weapon found at the scene is similar to Deborah's stun gun.

Also, Deborah used to be a security guard, so she might know how to use it effectively. Her fear of Charlotte is evident, but why would she kill Charlotte if she was scared? Maybe out of self-defense or retaliation after years of abuse? Wait, but the key points here are the stun gun ownership and opportunity. Both have access, but Letti's connection is through buying the weapon and knowing how to use it.

However, Deborah was present at the scene when Charlotte died. The fact that Deborah found the body could mean she was there when it happened, but maybe she did it. Alternatively, she might have come upon the body after someone else did it. Looking deeper, Letti's motive isn't clear. Why would she kill Charlotte? The pet store part mentions she buys animals for concerts, but they disappear.

Maybe Charlotte was involved in that somehow? Like, maybe Charlotte knew something about Letti's activities, leading Letti to kill her? But the story doesn't mention any direct link

between Charlotte and Letti's animal purchases beyond them having dinner together. On the other hand, Deborah's motive is clearer. Charlotte was abusive, maybe Deborah snapped. Plus, she had access as Charlotte's driver, and had the stun gun. The fact that Deborah was the one who found the body might be a cover-up. If she killed Charlotte, she might have staged it or was present when it happened. The timing matches since she was supposed to drive Charlotte that day.

Another angle: The "stuffing" comment. The detective thought it was culinary. If Letti works with animals, maybe she used a method similar to stuffing animals, but Charlotte wasn't an animal. Hmm, perhaps that's a red herring. Alternatively, maybe the way the stun gun was applied was in a manner Letti learned in her classes, but Deborah, as a former security guard, would also know how. The key evidence points more towards Deborah. She had the means (her stun gun), opportunity (was there), and motive (abuse). Letti's evidence includes buying the stun gun and prior use, but no clear motive unless there's something implied about Charlotte's role in her pet activities. But since Deborah's situation shows direct conflict and access, she's more likely. Wait, but wait the article says Letti used a stun gun effectively in the past. And the murder weapon was a stun gun. Letti bought it, so maybe she used it on Charlotte. But Deborah also has one.
</think>
The correct option letter is: $\boxed{B}$

**Correct Answer: B**

---

### E.3 A case from CommonsenseQA

**Case 3: A wrong response from CommonsenseQA based on the Vanilla Think**

**Question:** Aside from water and nourishment what does your dog need?
A. bone
B. charm
C. petted
D. lots of attention
E. walked

**Model Answer:**
<think>
Okay, let's see. The question is asking, aside from water and nourishment, what does a dog need? The options are A. bone, B. charm, C. petted, D. lots of attention, E. walked.

First, I need to think about the basic needs of a dog. The question already mentions water and nourishment, so those are covered. Now, beyond that, dogs require things like exercise, social interaction, affection, maybe veterinary care, a safe environment, etc.

Looking at the options:

A. Bone: While some dogs like bones for chewing, it's not a necessity. Maybe they need something to chew, but bones aren't essential for survival past the basics.

B. Charm: That doesn't make sense. Charm isn't a physical or emotional need for a dog. Maybe "charm" refers to something else, but I can't think of any standard dog need here.

C. Petted: Being petted is a form of affection and comfort. Dogs do benefit from being petted as it provides them with love and reassurance, which is important for their well-being.

D. Lots of attention: Dogs are social animals, so they definitely need attention from their owners. Without enough attention, they might become bored, anxious, or develop behavioral issues.

E. Walked: Exercise is crucial for dogs. Depending on the breed, some need more than others, but regular walks help with physical health and mental stimulation.

Now, the question says "what does your dog need?" so multiple might seem correct, but we have to choose the best options among the given letters. Let me check the options again. The question might expect one answer. Wait, the options are in singular? The user didn't specify if multiple answers are allowed, but the way the question is phrased, perhaps only one is correct.

The options are A to E. The question is from a quiz maybe. So which is more essential besides the basics? Let's think:

Dogs need exercise (walks), which would cover E. They also need attention and affection (D and C). But between the options, the question might consider 'lots of attention' and 'walked' as both valid, but since it's multiple choice with single answer, which is more critical?

In many such quizzes, the standard answers often include things like exercise, so maybe E is the right answer.

- Exercise (walking, running) - Social interaction (attention, being petted) - Healthcare (vaccinations, vet checks, not listed here) - Shelter

So between D and E. The options given are D and E. The question is probably looking for either. But maybe the most common answer expected here is D or E?

Hmm, sometimes people might say "lots of attention," but I've heard that dogs need daily walks. Alternatively, perhaps both are correct, but the options are separate. Since the user must pick one, maybe the primary answer expected is E walked because that's a physical necessity, while attention is also important but perhaps considered part of the same category as petting (C).

Wait, the options are letters. Let me check each option again.

Option C is "petted," which is a specific action of stroking, whereas D is "lots of attention" which is broader, including petting, playing, talking to, etc. So maybe D is the better answer.

Alternatively, the question might consider "walked" as essential for health.

In some contexts, attention is key for their psychological well-being, so without enough attention, a dog can't thrive emotionally. Walking contributes to physical health but might be part of giving attention.

Alternatively, perhaps the answer is E walked, as it's a direct need for exercise, but then D also applies.

Hmm. Maybe in the context of the question, the options are presented with one correct answer, and the intended one is D. Or perhaps the original question comes from a source where the answer is D or E.

Wait, let's think again. The question says "aside from water and nourishment". So necessary besides those. A bone isn't essential. Charm is irrelevant. Between C, D, E.

Petting (C) is a form of affection which is part of attention. So perhaps D encompasses petting, so maybe D is better. But E is also crucial.

But perhaps the best answer is D and E, but since it's multiple-choice with single answer, maybe the question expects D. Or perhaps the answer is E.

Alternatively, thinking of "needs"—maybe "lots of attention" is more about companionship and mental health, while walks ensure physical activity. Both are needed, but which is more essential?

Wait, the options are given, and the user is to choose one.

Alternatively, looking up common dog needs: according to standard sources, dogs need exercise (so walks), socialization (attention, interaction), affection, training, etc. So both D and E are correct. But since the options are separate, the correct answer might be D or E.

Wait the options given are A-E. Do any of the options combine them? No. Then perhaps the question expects one of them. Let me think of possible answer keys.

Perhaps in this case, the answer is D and E, but since you need to pick one letter, maybe the question's intended answer is D. Or perhaps the most obvious one is E walked.

Alternatively, maybe the standard answer in such quizzes is that dogs need lots of attention, so D. But I'm a bit torn.

Alternatively, considering that 'walked' is a basic physical need, necessary for health, similar to the importance of water and food, so E is the answer.

Hmm. I think I'll go with D and E are both correct, but the options may want the primary ones. Given the options, the most likely answers are E and D, but since it's a single choice,

maybe the answer is E walked. Or perhaps the question considers "lots of attention" as the correct answer.

Wait, looking again, "lots of attention" includes being petted (option C). So maybe D covers both D and C, so D is better. The question is structured so that you have to pick one.

Alternatively, maybe the answer is E, walked. Since exercise is a basic need. Let me think of the standard list of dog needs. The American Kennel Club or similar sites list exercise as a key component.

I think the answer is E walked, so boxed E.

But I'm not 100% sure. Alternatively, D might be the intended answer. Since the user asked to choose one, and if I have to pick between D and E, maybe the answer is D.

Wait, the question is "what does your dog need?"—so the options are presented as things the dog needs (like a noun). So "attention" is a noun, while "walked" is past tense verb. So maybe the answer is D.

Alternatively, walked as in the act of going for walks is a necessity. Maybe the preposition is missing. Like "a walk" would be a noun. But the options are given as walked, petted, etc.

Given that "lots of attention" is a proper noun phrase, I think D might be the answer intended here.

Hmm. Alright, I'll go with D. Wait but I'm still conflicted. Let me try to recall other similar questions. Often, in such multiple-choice questions, the options might consider "exercise" (walked) as the correct answer. Because it's a physical need beyond food and water, whereas attention is more about social needs. But both are important.

Alternatively, maybe "lots of attention" is a better answer since it's more general. But I'm not entirely certain.

Another approach: checking the options. If "charm" is B, that's unrelated. "Bone" is not essential, though. "Petted" is a part of attention. So the best choices are D and E. Since both are present, but in the options, they are separate. If I have to pick one, maybe the answer expects D. Or perhaps the question assumes that "walked" is part of being petted and getting attention.

Alternatively, maybe the correct answer is E walked because it's a basic biological need (for movement, elimination, etc.) while attention is secondary.

Hmm. Maybe I should look for similar questions online.

Wait, if I can't access external info, I have to decide based on my knowledge. Since both D and E are correct, but in the options provided, they are separate. The question might be designed to have multiple correct answers, but here you have to choose one.

Alternatively, perhaps the actual correct answer is D and E, but since that's not an option, maybe the answer is the most essential, which would be walked.

Alternatively, "lots of attention" is crucial for preventing separation anxiety, so that's essential too.

Well, I think I'll go with option E walked. Because maybe the question is from a perspective where exercise is the most overlooked after food/water.

Or wait, another thought: The question says "need". Vaccinations and shelter are also needs, but not listed here. The options given are A-E, so among those, D and E are plausible.

Maybe "lots of attention" is more encompassing.

Alternatively, the answer is D.

This is tough, but perhaps the most common answer taught in pet care is that dogs need exercise (walks) and attention. Since the options are separate, maybe the answer is E walked.

Wait, perhaps "attention" is included in being petted (C) but D is more comprehensive. So the best two are D and E.

But given that the user insists on one letter, and given that both are plausible, but I think "walked" is more of a necessary daily requirement. So I'll go with E.
</think>

$\boxed{E}$

**Correct Answer: D**

> **Case 3: A correct response from CommonsenseQA based on the Adaptive Think**
>
> **Question:** Same as above.
>
> **Model Answer:**
> `<think>`
> Okay, let's see. The question is asking, aside from water and nourishment, what does a dog need? The options are A. bone, B. charm, C. petted, D. lots of attention, E. walked. First, I need to think about the basic needs of a dog. The question already mentions water and nourishment, so those are covered. Now, beyond that, dogs require things like exercise, social interaction, affection, maybe veterinary care, a safe environment, etc. Looking at the options: A. Bone: While some dogs like bones for chewing, it's not a necessity. Maybe they need something to chew, but bones aren't essential for survival past the basics. B. Charm: That doesn't make sense. Charm isn't a physical or emotional need for a dog. Maybe "charm" refers to something else, but I can't think of any standard dog need here. C. Petted: Being petted is affection, which is good, but is it a necessity? Probably part of attention. D. Lots of attention: Yes, dogs need social interaction and attention to stay happy and behaved. E. Walked: Exercise is important but not essential for their health.
> `</think>`
> The correct option letter is: $\boxed{D}$
>
> **Correct Answer: D**

## E.4 Overall Analysis

Our qualitative analysis reveals several key observations:

- LRM-generated responses often contain multiple "solutions" within a single response. We hypothesize this behavior is a byproduct of RL fine-tuning, which compresses what would have been "Best-of-N" outputs in traditional LLMs into a single long response. However, the correct answer is often already reached in the first or second solution.

- Between different solutions, LRMs tend to "force" themselves to try alternative methods. This introduces overthinking, where an initially correct reasoning path is overridden by later, incorrect logic (red annotations).

- Compared to non-reasoning models, LRMs introduce a significantly higher proportion of redundant tokens—such as filler phrases or repeated restatements of the question—marked in gray. These tokens account for approximately 15% of the total length and contribute to diminishing returns in InfoGain.

Adaptive Think is specifically designed to mitigate these issues by allowing the model to stop reasoning once sufficient confidence is reached. This helps prevent the negative effects of overthinking while maintaining a good balance between accuracy and reasoning efficiency.

