# OpenReview forum: "Think or Not? Exploring Thinking Efficiency in Large Reasoning Models via an Information-Theoretic Lens"
_NeurIPS.cc/2025/Conference — NeurIPS 2025 spotlight_

### Official Review · Reviewer_SvkP · 2025-06-21

**Clarity:** 3
**Significance:** 2
**Originality:** 2
**Rating:** 4
**Confidence:** 4

**Summary:**

This paper investigates the inefficiencies in the reasoning processes of Large Reasoning Models (LRMs). The authors propose two novel metrics, InfoBias and InfoGain, to quantify deviations from ideal reasoning paths and the incremental information contribution of each reasoning step, respectively. Empirical analyses reveal that longer reasoning chains often exhibit higher information bias and diminishing information gain, especially for incorrect answers. Building on these insights, the authors introduce an entropy-based Adaptive Think strategy that dynamically halts reasoning once confidence exceeds a tunable threshold. Experiments across six benchmarks demonstrate its effectiveness.

**Questions:**

The paper is well-written and clear that I have no obvious confusion or question. I will maintain the rating unless the author can address the two weaknesses I concerned.

**Ethical Concerns:**

["NO or VERY MINOR ethics concerns only"]

**Final Justification:**

The author response is acceptable for me on the potential issues presented for me.  I believe the paper could be bordline accept after the improvements promised by the authors.

**Limitations:**

The author acknowledges that their Adaptive Think Strategy method has limitations. Although the authors propose workarounds (e.g., tree-based search for math problem) to address its limitation, these introduce additional non-negligible computational overhead and may not generalize well beyond constrained settings.

**Quality:**

2

**Strengths And Weaknesses:**

Strengths:
1.	Novel Metrics for Reasoning Efficiency: The introduction of InfoBias and InfoGain provides a rigorous, information-theoretic framework to evaluate reasoning efficiency in LRMs. These metrics effectively capture semantic alignment and stepwise uncertainty reduction, offering valuable tools for future research.
2.	Clarity and Reproducibility: The paper is well-structured, with detailed descriptions of methodologies, experimental setups, and hyperparameters. The availability of code and data further enhances reproducibility and transparency.


Weaknesses:
1.	Potential Pitfalls in Statistical Methodology:
The paper analyzes the relationship between reasoning length and efficiency by grouping all samples into correct and incorrect categories and computing InfoBias and InfoGain at the dataset level. However, this approach may introduce confounding factors. For instance, if the dataset contains both extremely simple and extremely difficult questions, LRMs may generate short reasoning chains for easy questions (likely correct) and long reasoning chains for hard questions (likely incorrect due to inherent difficulty rather than reasoning inefficiency). In such cases, the observed correlation between longer reasoning and higher InfoBias or lower InfoGain may simply reflect task difficulty rather than reasoning inefficiency. Additionally, prior work (Do not think that much for 2+ 3=?) suggests that incorrect reasoning chains often contain redundant or repetitive tokens, which may artificially inflate InfoGain without meaningful semantic deviation. The authors do not analyze the composition of these redundant tokens or disentangle whether increased reasoning length is a cause or a symptom of incorrect reasoning. A more fine-grained analysis—such as controlling for question difficulty or distinguishing between informative and non-informative reasoning steps—would strengthen the validity of their conclusions.

2.	Practical Limitations of the Adaptive Think Strategy:
The proposed entropy-based Adaptive Think method relies on two key assumptions that limit its applicability:
-  Dependence on a Closed Answer Space: The method requires enumerating all possible candidate answers to compute entropy, which is infeasible for open-ended tasks (e.g., free-form generation). Although the authors propose workarounds (e.g., tree-based search for math problem in the appendix), these introduce additional computational overhead and may not generalize well beyond constrained settings.
- Computational Overhead from Entropy Estimation: At each reasoning step, the model must pause generation, perform an additional inference pass to estimate entropy over the answer space, and then decide whether to continue. This introduces non-trivial latency, particularly since modern generation pipelines are not designed for frequent switching between decoding and prefilling modes. The paper does not quantify this overhead, leaving open questions about the method’s efficiency in real-world deployment.

These limitations suggest that the Adaptive Think strategy may be restricted to narrow, well-defined tasks with limited answer spaces and may not truly reduce generation latency to open-ended domains.

---

> ### Author Rebuttal · Authors · 2025-07-31
>
> Dear Reviewer SvkP,
>
> We express our genuine gratitude for the insightful comments and valuable concerns.
> ***
> > **W1: Potential Pitfalls in Statistical Methodology**
>
> **1. InfoBias and InfoGain analysis may be confounded by question difficulty**
>
> **For InfoBias, we also took this issue into consideration.** For example, the difference in the distribution of correct and incorrect samples along the *average token* axis in Figure 2 could potentially be caused by differences in question difficulty. To mitigate this, we used the GSM8K dataset, in which all questions are of comparable difficulty (elementary-school-level math), thereby minimizing large variations in problem difficulty. When computing InfoBias, we also **normalize the scores by the number of tokens in each response**, yielding a per-token dependency measure that allows for fairer comparisons (as noted in Appendix C.2). As shown in Figure 2, the distribution gap between correct and incorrect samples on the *normd InfoBias* metric is significantly larger than the gap observed on *average token*—especially for the two reasoning models—indicating that **our metric effectively reflects the model’s reasoning efficiency rather than problem difficulty**.
>
> Additionally, as requested, we conducted further experiments on **MATH500, evaluating questions across different difficulty levels to better control for question difficulty**. The results are as follows:
>
> #### **Table 1: JS Divergence of the InfoBias Distributions for Correct and Incorrect Samples Across Different Difficulty Levels in the MATH500 Dataset (QwQ-32B).**
> Metric|Level-1|Level-2|Level-3|Level-4|Level-5
> -|-|-|-|-|-
> **Accuracy**|81.40%|76.67%|71.43%|69.53%|60.45%
> **Average Token**|1444.51|2039.04|2921.08|4058.89|6529.85
> **JS-Divergence on InfoBias**|0.2746|0.2906|0.3127|0.3175|0.3395
>
> We categorized the questions by difficulty and calculated the Jensen–Shannon divergence (base-2) between the distributions of correct and incorrect samples on the InfoBias. Experimental results show that, **even after controlling for question difficulty, the differences between the distributions remain significant across all difficulty levels (0.2746–0.3395)**. Interestingly, the JS divergence of the InfoBias distributions increases with problem difficulty. This may be because harder questions typically involve longer and more complex reasoning paths, where incorrect reasoning is more likely to introduce larger semantic deviations, thereby resulting in greater differences.
>
> As for InfoGain, the calculation is performed at the granularity of each reasoning step, measuring the incremental contribution of each step. Our goal is to compare the gain contribution between non-reasoning and reasoning modes, showing that the information-gain contribution rate in reasoning models is slower than in non-reasoning models. **Question difficulty does not affect the calculation of InfoGain.**
>
> **2. Lack of analysis on redundant tokens in incorrect reasoning chains**
>
> **In Appendix E, we provide several case studies where Vanilla Think produces incorrect answers while Adaptive Think yields correct ones.** For these examples, we annotate the reasoning trajectories by categorizing the information into three types: (1) reasoning steps that positively contribute to the correct answer (black), (2) redundant or irrelevant content such as repetition or unnecessary elaboration (gray), (3) misleading or incorrect steps that contribute to the final wrong answer (red).
>
> Our qualitative analysis reveals several key observations:
> - **LRM-generated responses often contain multiple “solutions” within a single response**, echoing findings from *Do not think that much for 2 + 3 = ?*. We hypothesize this behavior is a byproduct of RL fine-tuning, which compresses what would have been “Best-of-N” outputs in traditional LLMs into a single long response. However, the correct answer is often already reached in the first or second solution.
> - **Between different solutions, LRMs tend to “force” themselves to try alternative methods.** This introduces overthinking, where an initially correct reasoning path is overridden by later, incorrect logic (red annotations).
> - **Compared to non-reasoning models, LRMs introduce a significantly higher proportion of redundant tokens**—such as filler phrases or repeated restatements of the question—marked in gray. These tokens account for approximately 15% of the total length and contribute to diminishing returns in InfoGain.
>
> Adaptive Think is specifically designed to mitigate these issues by allowing the model to stop reasoning once sufficient confidence is reached. This helps prevent the negative effects of overthinking while maintaining a good balance between accuracy and reasoning efficiency.
>
> > **W2: Practical Limitations of the Adaptive Think Strategy**
>
> **1.	The method assumes a closed answer space, limiting its applicability to open-ended tasks**
>
> We acknowledge this limitation, which we have also discussed in Appendix B.1. **Our method is primarily designed for reasoning tasks, where a ground truth answer is available and the answer space can be enumerated or approximated—this aligns with the dominant application scenarios of current reasoning models.** To validate the applicability of Adaptive Think across different answer formats, we conducted experiments on both **open-ended QA tasks** (GSM8K, AIME2025) and **multiple-choice tasks** (MMLU-Pro, MuSR, ProntoQA, and CommonsenseQA).
>
> **For the open-ended tasks of particular interest to the reviewer, GSM8K and AIME2025 involve free-form generation.** While the answers are relatively short, our experiments demonstrate that Adaptive Think effectively improves reasoning efficiency in these settings. To reduce the compute overhead of tree-based search, we approximate the answer distribution using the beam search, sampling only a fixed number of candidate answers. This provides a tractable approximation without exhaustive enumeration. **A detailed quantitative analysis is provided in the following response.**
>
> For longer-form generation tasks, we agree that extending Adaptive Think to such settings poses a meaningful challenge and represents a promising direction for future work in areas such as creative writing or summarization.
>
> **2.	Entropy estimation introduces computational overhead not quantified in the paper**
>
> We thank the reviewer for raising the concern regarding potential compute overhead from entropy estimation. First, our method operates entirely **within the standard decoding process**. After generating each reasoning step, we append a short, fixed answer prompt and compute the probability distribution over the candidate answers. This is a lightweight forward pass that continues from the existing decoding state — it does not require reconstruction of the KV cache. **Therefore, our approach does not involve switching between decoding and prefilling modes, and does not incur any system-level latency from mode transitions.**
>
> Next, we provide a rigorous analysis of the computational overhead introduced by the adaptive think strategy. We define: $C_m$ (Compute cost per token), $T$ (Average tokens per reasoning step).
>
> - **Vanilla Think**: $C_{vanilla}=C_mL_v=C_mS_vT$ (number of steps: $S_v$, total tokens: $L_v=S_vT$).
> - **Adaptive Think**: $S_a$ (early stopping steps), $L_a=S_aT$ (total tokens), and $r_t=1-\frac{L_a}{L_v}=1-\frac{S_a}{S_v}$ (token reduction).
>     - **Multiple-Choice Tasks**: $C_{MC}=C_mS_aT+S_aC_e$ (generation+overhead). $C_e$ is the cost of extracting probabilities and computing entropy. Compared to the vanilla method, we have $C_{MC}<C_{vanilla}\iff S_aT+\frac{S_aC_e}{C_m}<S_vT$. Substituting $C_e/C_m\approx 10^{-3}$ (scalar ops are $10^3-10^4$× faster than matrix ops on GPUs) and $S_a/S_v=1-r_t$, we have $\frac{S_aC_e}{C_m}\ll T$, thus the overhead is negligible and savings are approximately equal to token reduction rate $r_t$.
>     - **Free-Form Generation Tasks**: $C_{FF}\approx C_mS_aT+S_a(C_mWN)=C_mS_a(T+WN)$, where $N$ is the tree search depth and $W$ is the beam width. Notably, other overheads such as path probability and entropy calculations are constant level complexity, which is negligible compared to $C_m$, so we only focus on the additional overhead caused by tree search. Compared to the vanilla method, we have $C_{FF}<C_{vanilla}\iff S_a(T+WN)<S_vT\iff \frac{S_a}{S_v}<\frac{T}{T+WN}$, which means that the reasoning efficiency is improved if this formula is satisfied. In our experiment, we control the minimum thinking step length of $120$, tree search depth $N=10$, and beam width $W=5$. Therefore, in the extreme case (tree searches are performed every shortest length $T=120$, and each search reaches the deepest), the token reduction rate is considered to be satisfied when the token reduction rate is greater than $1-120/(120+5*10)=29.41$%. Our average token reduction on the two free-form generated datasets was $58.78$%, significantly reducing generation latency.
>
> The above analysis demonstrates that Adaptive Think can effectively optimize reasoning efficiency during the model’s decoding stage for both multiple-choice tasks and free-form generation tasks. **Although the tree search process introduces additional computational overhead, it still achieves a certain level of improvement compared to Vanilla Think.** Moreover, the depth and beam width of the tree search can be flexibly adjusted based on the task’s response requirements, providing a degree of adaptability.
>
> ***
> We sincerely appreciate your thoughtful comments and suggestions. We are committed to continuously improving this work. Please do let us know if our response has sufficiently addressed your concerns.

---

> > ### Comment · Reviewer_SvkP · 2025-08-05
> >
> > Thanks for the response.  It has addressed all my concerns.

---

> > > ### Author Response · Authors · 2025-08-05
> > >
> > > Thank you sincerely for your valuable and constructive feedback on our work! We will incorporate your suggestions in the next version by **including more detailed analysis on question difficulty control and computational overhead**, as well as **expanding the case studies and experimental results in the appendix**. Your insights have significantly strengthened the clarity and rigor of our paper.
> > >
> > > We truly appreciate your time and thoughtful review, and we hope our responses have sufficiently addressed your concerns to **merit a favorable final evaluation**.

---

### Official Review · Reviewer_3P8b · 2025-06-29

**Clarity:** 3
**Significance:** 3
**Originality:** 4
**Rating:** 5
**Confidence:** 4

**Summary:**

This paper investigates the trade-off between reasoning length and efficiency in Large Reasoning Models (LRMs) from an information-theoretic perspective. The authors introduce two metrics—InfoBias and InfoGain—to quantify inefficiencies and diminishing returns in extended reasoning chains. Building on these insights, they propose a training-free, entropy-based Adaptive Think strategy that dynamically halts reasoning when the model’s confidence is sufficiently high. Experimental results show that this approach improves accuracy by 1.10% and reduces token usage by 50.80% compared to the default reasoning mode, demonstrating improved efficiency across diverse benchmarks.

**Questions:**

See "weaknesses"

**Ethical Concerns:**

["NO or VERY MINOR ethics concerns only"]

**Final Justification:**

The rebuttal addresses my concerns well, and I will keep my scores.

**Limitations:**

yes

**Quality:**

4

**Strengths And Weaknesses:**

**Strengths**

1. The research problem is highly relevant, particularly for optimizing test-time computation.
2. The paper is well-structured, clearly presenting the problem, analyzing it, and offering a practical solution.
3. The proposed method is training-free, making it highly accessible and valuable to the broader community.

**Weaknesses**

1. While the concepts of InfoBias and InfoGain are understandable from the LLM perspective, the connection between information-theoretic principles and LLM entropy could be explained more clearly for readers unfamiliar with this domain.
2. It would be interesting to explore whether this strategy could also be used to guide model training, enabling LLMs to learn adaptive reasoning behaviors.
3. Additional results on smaller models (e.g., Qwen-DeepSeek-7B) would help assess the generalizability and effectiveness of the proposed strategy across model scales.

---

> ### Author Rebuttal · Authors · 2025-07-31
>
> Dear Reviewer 3P8b,
>
> We express our genuine gratitude for the insightful comments and valuable concerns.
> ***
> > **W1: The connection between information-theoretic concepts and LLM entropy needs clearer explanation**
>
> Indeed, in our paper, we use concepts such as entropy, information gain (InfoGain), and information bias (InfoBias) to analyze reasoning behaviors in language models. **These metrics are not merely analogies—they have direct structural correspondences with principles in classical information theory.**
>
> From an information-theoretic perspective, entropy quantifies uncertainty. In language models, the token-level probability distribution represents the model’s uncertainty about “what the next token will be.” This aligns precisely with Shannon’s definition of information entropy. **Thus, the model’s prediction distribution at any point in time can be viewed as the output of an information source, and the entropy of that distribution reflects its current uncertainty.** Building on this foundation, we further define: **1) InfoGain**: Each reasoning step $s_t$ alters the model’s belief about the answer distribution. This is analogous to the classical notion of “uncertainty reduction after observing an event.” Each reasoning step can be seen as an intermediate process in compressing uncertainty during information transmission. **2) InfoBias**: We use InfoBias to measure whether the model’s reasoning process exhibits a deviation from the correct reasoning trajectory—that is, whether it is being misled by incorrect “information guidance.” This concept is motivated by information-theoretic analyses of channel bias, where bias in the transmission channel leads to misinterpretation or distortion of the received information.
>
> In summary, the metrics we propose are firmly grounded in Shannon’s definition of entropy and are directly mapped to the response- and step-level prediction dynamics of language models. We argue that viewing the reasoning process in LLMs as one of information compression and transmission provides a systematic framework for understanding and optimizing model behavior.
>
> **We will include these formal definitions and their theoretical foundations in the revised version of the paper, along with illustrative figures and examples in the appendix to help readers intuitively grasp the explanatory power of this perspective.** We sincerely thank the reviewer for this suggestion, which has helped us better communicate the relevance and value of the information-theoretic lens to a broader interdisciplinary audience.
>
> > **W2: The potential of using the strategy to guide model training**
>
> We thank the reviewer for highlighting this forward-looking and important direction. **The core objective of our work is to serve as a preliminary exploration into how InfoBias and InfoGain can help us better understand the reasoning behavior of current LRMs.** We observe that mainstream LRMs, when tackling reasoning tasks, often produce overly lengthy chains of thought, filled with redundant content, ineffective repetition, or even misleading steps. This phenomenon of “overthinking” not only undermines reasoning efficiency but may also reduce overall accuracy.
>
> Against this backdrop, we propose a lightweight entropy-based strategy—*Adaptive Think*—which dynamically monitors changes in the model’s confidence toward the final answer during inference. Based on this, the method decides whether to continue reasoning. Our aim is to validate whether information-theoretic signals can effectively characterize and regulate the reasoning process in models.
>
> Regarding potential applications during the training stage, we fully agree with the reviewer’s insight: incorporating InfoBias or InfoGain as auxiliary supervision or reward signals could potentially enhance LRM’s reasoning efficiency and decision-making capabilities. For example:
> - During **supervised fine-tuning**, regularization terms based on effective reasoning length or entropy convergence speed could be introduced;
> - In **reinforcement learning**, rewards could be designed to encourage maximizing information gain or minimizing redundancy;
> - Models could be guided to learn **when to terminate generation**, enabling truly self-monitoring reasoning.
>
> Although there is already some emerging work focusing on improving the reasoning efficiency of models, we believe that information theory offers a more principled framework—providing models with a form of intrinsic reasoning supervision. In our view, this is a highly promising direction and will be a major focus of our future research.
>
> > **W3: Lack of results on smaller models limits assessment of generalizability across model scales**
>
> Considering that small-scale models (e.g., 7B) would severely limit the reasoning capabilities of LRMs—potentially leading to “pseudo-thinking” behavior—this is particularly true for models like DeepSeek-R1-Distill, which are trained via distillation rather than reinforcement learning from scratch. Moreover, selecting the 32B scale allows us to conduct experiments on both QwQ and DeepSeek-R1-Distill, thereby increasing model diversity, as QwQ does not have smaller-scale versions available. For these reasons, we initially chose to experiment only with 32B reasoning models.
>
> We greatly appreciate your suggestion. **To further assess the generalizability and effectiveness of the proposed strategy across model scales, we additionally conducted experiments on DeepSeek-R1-Distill-Qwen-7B**:
>
> #### **Table 1: Results of DeepSeek-R1-Distill-Qwen-7B on Six Reasoning Datasets**
> Think Mode|GSM8K (Acc / #Token)|AIME2025 (Acc / #Token)|MMLU-Pro (Acc / #Token)|MuSR (Acc / #Token)|ProntoQA (Acc / #Token)|CommonsenseQA (Acc / #Token)
> -|-|-|-|-|-|-
> **Vanilla Think**|85.47% / 433.08|45.33% / 10885.24|57.43% / 1287.57|39.79% / 1036.89|93.60% / 1082.80|62.95% / 481.40
> **No-Think**|87.85% / 264.79|8.67% / 957.08|40.29% / 223.87|37.96% / 97.29|76.80% / 372.69|53.66% / 62.98
> **Gated Think**|87.34% / 279.76|26.67% / 5958.27|41.43% / 234.59|38.62% / 172.63|72.00% / 406.34| 52.19% / 89.36
> **Adaptive Think**|**87.95% / 259.80**|**47.33% / 4188.40**|**58.57% / 722.84**|**41.40% / 725.35**|**93.67% / 979.80**|**63.02% / 266.98**
> **Delta vs. Vanilla**|+2.90% / -40.01%|+4.41% / -61.52%|+1.99% / -43.86%|+4.05% / -30.05%|+0.07% / -9.51%|+0.11% / -44.54%
>
> Experiments show that the *Adaptive Think* improved reasoning efficiency for DeepSeek-R1-Distill-Qwen-7B across all six datasets, **reducing token usage by an average of 38.25% compared to *Vanilla Think***. Notably, even on this smaller-scale model, **accuracy across different types of reasoning tasks also increased slightly, with an average improvement of 2.26%**.
>
> These results further demonstrate that *Adaptive Think* can significantly enhance reasoning efficiency not only across different model types (QwQ and DeepSeek) but also across different model scales (32B and 7B), while maintaining or even improving accuracy.
>
> ***
> We sincerely appreciate your thoughtful comments and suggestions. We are committed to continuously improving this work. Please do let us know if our response has sufficiently addressed your concerns.

---

> > ### Comment · Reviewer_3P8b · 2025-08-04
> >
> > Thanks for the clarification, which addresses my concerns well. I will keep my scores and look forward to the revision!

---

> > > ### Author Response · Authors · 2025-08-05
> > >
> > > Thank you once again for dedicating your time and effort to reviewing our work. We will certainly incorporate your valuable suggestions into our revision!

---

### Official Review · Reviewer_uc9j · 2025-07-02

**Clarity:** 3
**Significance:** 3
**Originality:** 3
**Rating:** 5
**Confidence:** 3

**Summary:**

This paper systematically investigates the efficiency of current large reasoning language models (LRMs) during inference from an information-theoretic perspective, focusing on how long reasoning chains under various task settings cause the accumulation of information bias and introduce semantic redundancy. By introducing two metrics—InfoBias and InfoGain—the authors show that over-reasoning tends to add semantic redundancy while offering limited gains.

Building on this insight, they propose an entropy-based strategy called Adaptive Think that dynamically adjusts the length of reasoning chains according to confidence thresholds, matching the depth of thought to task difficulty and sharply reducing unnecessary steps. Experiments on diverse reasoning benchmarks demonstrate that this approach significantly lowers average token usage while maintaining or improving final accuracy, validating information-theoretic metrics as effective tools for monitoring and optimizing LRM reasoning workflows across different tasks.

**Questions:**

Please address the weaknesses above.

**Ethical Concerns:**

["NO or VERY MINOR ethics concerns only"]

**Final Justification:**

Thanks for the response! I believe my main concerns are addressed. Still, I believe that more empirical evaluation on other datasets is necessary. Only evaluating on GSM8K is not sufficient. So I recommend that the authors add more experiments in the revised version.

**Limitations:**

Yes

**Quality:**

3

**Strengths And Weaknesses:**

Strength:

1. The paper redefines reasoning efficiency from an information-theoretic standpoint (entropy and mutual information), introducing two metrics—InfoBias and InfoGain—that offer valuable insights for analyzing how LRMs reason across different tasks.

2. The proposed theory is systematically validated on multiple, diverse benchmarks, giving the conclusions high credibility. Figure 3 clearly illustrates how entropy and probability evolve throughout the model’s reasoning process, leading to several valuable findings. Follow-up quantitative experiments further confirm the method’s effectiveness: Adaptive Think saves 40–80 % of tokens while slightly improving accuracy.

3. The entropy-based Adaptive Think strategy is simple and straightforward, can be seamlessly integrated into existing LLM inference pipelines, and is practically valuable for reducing computational costs and improving reasoning efficiency in real-world deployments.


Weaknesses:

1. Figure 2 only presents results for multiple models on the GSM8K dataset. Are there results for any other datasets?

2. Existing work—such as Dynasor-CoT [1]—also seeks to improve reasoning efficiency and reduce token usage. The authors can 1). clarify the key distinctions between Dynasor-CoT and Adaptive Thinking; 2). highlight any empirical advantages their method offers (e.g., fewer tokens, faster inference, or higher accuracy).

[1]Fu, Yichao, et al. "Reasoning without self-doubt: More efficient chain-of-thought through certainty probing." ICLR 2025 Workshop on Foundation Models in the Wild. 2025.

---

> ### Author Rebuttal · Authors · 2025-07-31
>
> Dear Reviewer uc9j,
>
> Thank you for the constructive comments. We provide our feedbacks as follows.
> ***
> > **W1: Results in Figure 2 are limited to GSM8K, with no evaluation on other datasets**
>
> To compute the InfoBias of the reasoning model outputs, we employed a mutual information–based calculation method, comparing the generated reasoning with the ideal path. This approach requires access to accurately annotated reasoning steps. **Among all the datasets we considered, only GSM8K provides ground-truth solutions; therefore, we conducted our initial analysis experiments using this dataset.**
>
> We also considered conducting experiments on additional datasets—for instance, MATH500, which also provides correct solution processes. However, **to control for the potential confounding effect of problem difficulty on the final analysis metrics (e.g., models may produce shorter reasoning chains for easier problems and longer ones for harder problems, causing InfoBias to reflect problem difficulty rather than true reasoning efficiency)**, we ultimately chose to perform our analysis experiments on GSM8K, which contains questions of more balanced difficulty levels, rather than the MATH500 dataset, which spans multiple difficulty levels.
>
> We appreciate your interest in this aspect of our analysis. In future work, we will consider identifying other suitable datasets to make the datasets used for our reasoning-efficiency analysis more diverse.
>
> > **W2: Lack of comparison with *Dynasor-CoT* in terms of method and empirical performance**
>
> Thank you for sharing the reference work *Dynasor-CoT*. It indeed proposes an inspiring mechanism for improving reasoning efficiency. This work identifies a notable issue of “self-doubt” in current LRMs, where the model repeatedly verifies answers despite already arriving at the correct one, leading to unnecessary token consumption. To address this, this work introduces the *Probe-In-The-Middle* technique, which inserts prompts mid-inference to extract intermediate answers in real time, thereby revealing the model’s “internal confidence” state. Without requiring additional training, they combine answer consistency checks and uncertainty indicators (such as “wait” or “hmm”) to implement an effective early termination strategy.
>
> While our shared goal is to enhance the reasoning efficiency of LRMs and reduce token usage, our method differs fundamentally from *Dynasor-CoT* and offers distinct empirical advantages in the following aspects:
>
> 1. **Differences in Termination Decision Criteria**
>
>     Although both *Dynasor-CoT* and our method, *Adaptive Think*, aim to assess the model’s “confidence level” to determine whether further reasoning is necessary, there are essential differences in how this “confidence” is modeled—leading to distinct signal sources, granularity, and discriminative capabilities in the termination strategies.
>     - *Dynasor-CoT* focuses on **output answer consistency**. It periodically inserts probe prompts during the reasoning process to elicit the model’s current “Final Answer” explicitly, and compares these intermediate outputs across different stages. If the model repeatedly produces the same answer at multiple probe points, it is deemed to be “sufficiently confident,” prompting early termination. This approach infers internal confidence based on consistency in surface-level outputs—an external and relatively coarse-grained signal.
>     - In contrast, *Adaptive Think* deeply models the **informational dynamics of the reasoning process**, quantifying the contribution of each reasoning step by directly measuring the entropy reduction and confidence gain within the latent answer space. Rather than comparing explicit answers post hoc, we continuously track the internal uncertainty resolution process as reasoning unfolds. This allows our method to:
>         - Detect the model’s confidence at an earlier stage;
>         - Identify with finer granularity whether the reasoning is still contributing meaningfully or if information gain is plateauing;
>         - Provide a unified and interpretable termination signal based on information gain or entropy, rather than heuristic rules.
>
>     Moreover, as shown in §3, our *InfoGain* analysis reveals that many models rapidly converge to the correct answer distribution in the early stages of reasoning, with subsequent steps offering diminishing—sometimes even negative—information gain. These findings offer strong theoretical and empirical support for our approach.
>
>     In summary, *Dynasor-CoT* relies on **external behavior signals** based on output consistency, whereas our method is rooted in **internal semantic entropy dynamics**, offering deeper insight and greater discriminative power in assessing the model’s cognitive state.
> 2. **Comparison of Reasoning Granularity and Semantic Completeness**
>     - Our approach adopts a granularity for termination decisions that better aligns with the semantic and logical structure of reasoning. Specifically, *Adaptive Think* operates at the semantic chunk level—evaluating whether to continue reasoning only after completing a full reasoning step. At each such step, the decision is based on the entropy of the current answer distribution. By segmenting the process according to natural-language paragraph boundaries, each decision point corresponds to a logically complete reasoning unit.
>     - In contrast, *Dynasor-CoT* uses a fixed-token interval strategy—e.g., inserting probes every 32 or 64 tokens. This token-based truncation can easily disrupt semantic boundaries, triggering premature assessments before a reasoning step is fully expressed. Such interruption often leads to distorted interpretations of the model’s internal state. For instance, in math or logic tasks, a single coherent reasoning step may span dozens of tokens, and mid-step probing can yield fragmented or incoherent outputs.
>
>     As detailed in our information-theoretic analysis (§3.3–3.4), we use the dynamic trend of information gain to quantify the actual contribution of each reasoning step. This analysis further confirms the critical importance of reasoning granularity and the timing of termination decisions. Information gain across steps is highly nonlinear, with certain key chunks contributing disproportionately to model confidence. Evaluating entropy at these natural semantic boundaries—rather than applying uniform probing intervals—proves more effective in capturing truly meaningful termination signals.
>
> 3. **Final Performance Comparison: Efficiency, Accuracy, and Task Adaptability**
>
>     In terms of overall performance, our method demonstrates superior results across several key dimensions:
>
>     - **Significantly Higher Token Savings**: While *Dynasor-CoT* has shown commendable token savings in mathematical tasks (e.g., up to 29% reduction on AIME and MATH500), *Adaptive Think* achieves substantially greater compression across multiple benchmarks. For example, we reduce token usage by 66.46% on GSM8K and 68.25% on AIME2025. This substantial improvement stems from our entropy-based fine-grained control, which assesses at each step whether further reasoning is “worth continuing.”
>     - **Accuracy Improves, Rather than Suffers**: Unlike most early stopping methods, which often face a trade-off between reducing tokens and preserving accuracy, *Adaptive Think* exhibits a gain—achieving higher accuracy with fewer tokens. This suggests our termination mechanism not only conserves computational resources but also suppresses semantic drift and overthinking induced by redundant reasoning.
>     - **Stronger Generalization and Tunability Across Tasks**: While *Dynasor-CoT* primarily focuses on mathematical tasks (e.g., AIME, AMC, MATH500), *Adaptive Think* has been systematically evaluated across six categories of reasoning tasks: Math (GSM8K, AIME2025), Logic (MuSR), Knowledge-intensive QA (MMLU-Pro), Binary judgment tasks (ProntoQA), and Commonsense reasoning (CommonsenseQA). This broad evaluation demonstrates the robustness and versatility of our approach across domains.
>     A core advantage of our method lies in its adaptability. By adjusting the control parameter α (entropy threshold coefficient), the system can dynamically adapt to varying reasoning depths required by different tasks:
>         - For **knowledge-intensive or structurally complex tasks** (e.g., MMLU-Pro, ProntoQA), a lower α ensures deeper, more thorough reasoning before stopping.
>         - For **intuitive or shallow tasks** (e.g., CommonsenseQA), a higher α enables early termination of redundant steps, achieving maximal compression without compromising performance.
>
>         We demonstrate this behavior experimentally: as α varies, the model achieves optimal **accuracy–token trade-offs** on different tasks (see Figure 6). This level of **task-aware tunability** is a direct result of our entropy-based decision mechanism—something that probe-based consistency methods like *Dynasor-CoT* lack the flexibility to support.
> ***
> We sincerely appreciate your thoughtful comments and suggestions. We are committed to continuously improving this work. Please do let us know if our response has sufficiently addressed your concerns.

---

### Official Review · Reviewer_5Tnb · 2025-07-03

**Clarity:** 3
**Significance:** 3
**Originality:** 3
**Rating:** 4
**Confidence:** 3

**Summary:**

This paper aimed to mitigate the redundant issue of long CoT in large reasoning models, and maintain the reasoning performance. Concretely, this work first analyzed the reasoning processes with the proposed InfoBias and InfoGain, finding some interesting patterns that related to the prediction accuracy. Based on the findings, the authors presented an entropy-based adaptive think strategy to reduce the LongCoT budgets and reserve the accuracy. Empirical results demonstrated the effectiveness for some models and tasks of the proposed method.

**Questions:**

1. During the investigation, the authors focused on the GSM8k, all questions are simple. Why did not use the MATH500 that contains questions of different difficulties? Analysis on MATH500 would be better than GSM8k.
2. How was the upper bound of the information bias computed? The derivation of the upper bound of Eq (2) needs more details.

**Ethical Concerns:**

["NO or VERY MINOR ethics concerns only"]

**Final Justification:**

I feel satisfied with the rebuttal and will keep my scores.

**Limitations:**

yes

**Quality:**

3

**Strengths And Weaknesses:**

**Strengths:**
1. This paper designed 2 information theory-based metrics and employed them to analyze the LongCoT processes, finding some interesting phenomena that are in line with human intuition.
2. Inspired by the analysis results, the authors present appropriate test-time strategy to decrease the reasoning tokens and keep prior performance, and experimental results proved the efficiency.
3. Experiments on different models and benchmarks illustrated that the proposed adaptive-think can beat existing think paradigms on reducing LongCoT tokens in most math reasoning scenarios.

**Weaknesses:**
1. This paper employed 32B models (QwQ, R1-distill-32B) for reasoning while relative small-scale models (Llama3.1-8B-instruct, Qwen2.5-7B-Instruct, and Phi-4 (14B)) for non-reasoning, such comparisons are not consistent and fair, undermining the faithfulness of the experimental conclusions.
2. In Section 3.1-3.3, the segmentation, ideal reasoning trajectory annotation, and step splitting can impact the later analysis significantly, however, the accuracy and authority of these two procedures were not elaborated thoroughly to make it convincible, leading to the later analysis unstable.
3. The proposed strategy performed well on math reasoning tasks, while unstable on other tasks as shown in Table 2 and Figure 6, implying the adaptive-think might lack generalization capability for broader domains.

---

> ### Author Rebuttal · Authors · 2025-07-31
>
> Dear Reviewer 5Tnb,
>
> We are deeply appreciative of your insightful and constructive feedback.
> ***
> > **W1: Use of Larger Models for Reasoning vs. Smaller Models for Non-Reasoning Leads to Unfair Comparison**
>
> In our experiments, we compared Base vs. CoT for non-reasoning models to show that longer reasoning doesn’t always improve accuracy, and Vanilla Think vs. No-Think vs. Gated Think for reasoning models to highlight inefficiencies in current LRM reasoning. **To better control for variables, we included results from non-reasoning models of comparable scale** (such as Qwen2.5-32B-Instruct and Yi-1.5-34B-Chat) **as well as smaller reasoning models** (e.g., DeepSeek-R1-Distill-Qwen-7B).
> #### **Table 1: Performance Results of Additional Non-Reasoning and Reasoning Models at Different Scales.**
> Model|Think Mode|GSM8K (Acc / #Token)|AIME2025 (Acc / #Token)|MMLU-Pro (Acc / #Token)|MuSR (Acc / #Token)|ProntoQA (Acc / #Token)|CommonsenseQA (Acc / #Token)
> -|-|-|-|-|-|-|-
> **Yi-1.5-34B-Chat**|Base|82.73% / 310.93|0.00% / 646.04|51.43% / 173.11|40.26% / 112.16|87.20% / 363.33|71.20% / 84.74
> **Yi-1.5-34B-Chat**|CoT|82.18% / 313.95|0.00% / 919.05|54.29% / 444.46|42.78% / 358.38|91.07% / 433.95|68.31% / 241.79
> **Qwen2.5-32B-Instruct**|Base|95.24% / 292.68|10.00% / 797.88|75.71% / 188.69|44.23% / 144.97|99.87% / 222.85|86.17% / 76.63
> **Qwen2.5-32B-Instruct**|CoT|94.63% / 304.26|10.67% / 830.06|80.86% / 375.51|46.51% / 302.17|99.73% / 276.56|85.36% / 203.74
> **DeepSeek-R1-Distill-Qwen-7B**|Vanilla Think|85.47% / 433.08|45.33% / 10885.24|57.43% / 1287.57|39.79% / 1036.89|93.60% / 1082.80|62.95% / 481.40
> **DeepSeek-R1-Distill-Qwen-7B**|No-Think|87.85% / 264.79|8.67% / 957.08|40.29% / 223.87|37.96% / 97.29|76.80% / 372.69|53.66% / 62.98
> **DeepSeek-R1-Distill-Qwen-7B**|Gated Think|87.34% / 279.76|26.67% / 5958.27|41.43% / 234.59|38.62% / 172.63|72.00% / 406.34|52.19% / 89.36
> **DeepSeek-R1-Distill-Qwen-7B**|Adaptive Think|87.95% / 259.80|47.33% / 4188.40|58.57% / 722.84|41.40% / 725.35|93.67% / 979.80|63.02% / 266.98
> **DeepSeek-R1-Distill-Qwen-7B**|Delta vs. Vanilla|+2.90% / -40.01%|+4.41% / -61.52%|+1.99% / -43.86%|+4.05% / -30.05%|+0.07% / -9.51%|+0.11% / -44.54%
>
> It is important to note that our primary focus is to explore ways to improve the reasoning efficiency of reasoning models. Therefore, our subsequent experiments mainly compare different reasoning strategies using 32B-scale reasoning models. We did not choose smaller reasoning models for the following reasons:
> - **The 32B scale allows us to conduct experiments on both QwQ and DeepSeek-R1-Distill, increasing model diversity.** QwQ does not provide smaller model variants.
> - **Smaller models such as 7B severely limit reasoning capabilities and may exhibit “pseudo reasoning” behavior**, especially in models like DeepSeek-R1-Distill, which are trained via distillation rather than reinforcement learning from scratch.
>
> > **W2: Lack of Detailed Explanation for Segmentation, Trajectory Annotation, and Step Splitting Procedures**
>
> Thanks for pointing this out. Trajectory annotation is crucial for the computation and analysis of InfoBias, and the choice of reasoning process segmentation method is equally important for InfoGain. **In fact, we conducted extensive research and preliminary experiments before settling on the current approaches.**
>
> Specifically, for **trajectory annotation**, using **LLM-generated** reasoning paths introduces the model’s own bias, which can distort mutual information estimates. Therefore, we opted for the more widely accepted **human-annotated** data. Since datasets containing human-written reasoning paths are limited, we conducted our analysis experiments primarily on GSM8K.
>
> For **reasoning process segmentation**, we experimented with several methods: **1) segmentation based on length or sentence boundaries; 2) LLM-assisted segmentation; and 3) paragraph-based segmentation.** Length- or sentence-based segmentation often leads to computing information gain on incomplete semantic units or reasoning steps, which distorts the answer distribution and causes unstable gain curves. LLM-assisted segmentation, on the other hand, requires an additional LLM to decide segment boundaries, significantly undermining the inference efficiency during decoding. Considering these factors, we ultimately chose paragraph-based segmentation. Empirically, we observed that the end of a paragraph often coincides with the completion of a reasoning step, which aligns well with our segmentation objective.
>
> > **W3: Proposed Strategy Performs Well on Math Tasks but Shows Instability on Other Domains**
>
> The primary goal of our work is to **improve the reasoning efficiency** of language models. As such, we focus on reducing token usage across a wide range of domain-specific datasets. Experimental results confirm that our proposed method achieves substantial token reduction across various types of reasoning tasks—including multi-step numerical reasoning, symbolic and abstract reasoning, knowledge-intensive reasoning, and narrative comprehension. Specifically, on the QwQ-32B model, our method reduces token usage by 24.37% to 68.25%, with an average reduction of 50.80% across tasks.
>
> At the same time, we maintain high reasoning accuracy. Among eight experiments conducted on non-mathematical datasets, only two showed a minor drop in accuracy (1.55%–2.11%), which remains within an acceptable range, while the remaining six tasks saw slight accuracy improvements.
>
> These results demonstrate that **our method significantly enhances reasoning efficiency without compromising accuracy, achieving robust performance across diverse task types.**
>
> > **Q1: Evaluation Limited to GSM8K; MATH500 Would Offer Better Diversity in Question Difficulty**
>
> In our initial analysis of reasoning efficiency, although both GSM8K and MATH500 provide annotated ground-truth reasoning paths, we chose to use GSM8K, which features questions of more uniform difficulty, in order to **control for the potential confounding effect of problem difficulty on the InfoBias metric**. For example, models may naturally produce shorter reasoning chains for easier problems and longer ones for harder ones—causing InfoBias to reflect question difficulty rather than true reasoning inefficiency. In the subsequent experiments, we therefore continued to validate our method on the GSM8K dataset.
>
> We sincerely thank the reviewer for suggesting a new perspective—**analyzing Adaptive Think across different levels of question difficulty**. In response, we have now included additional experiments on MATH500 to further examine this aspect：
> #### **Table 2: Results of QwQ-32B on the MATH500 Dataset.**
> Think Mode|Total (Acc / #Token)|Level-1 (Acc / #Token)|Level-2 (Acc / #Token)|Level-3 (Acc / #Token)|Level-4 (Acc / #Token)|Level-5 (Acc / #Token)
> -|-|-|-|-|-|-
> **Vanilla Think**|69.80% / 3893.76|81.40% / 1444.51|76.67% / 2039.04|71.43% / 2921.08|69.53% / 4058.89|60.45% / 6529.85
> **Adaptive Think**|70.80% / 1506.31|83.72% / 648.56|78.89% / 793.60|73.33% / 1087.34|68.75% / 1412.85|61.19% / 2677.82
> **Delta vs. Vanilla**|+1.43% / -61.31%|+2.85% / -55.10%|+2.90% / -61.08%|+2.66% / -62.78%|-1.12% / -65.19%|+1.22% / -59.00%
>
> Experimental results show that **Adaptive Think not only significantly reduces reasoning length across most difficulty levels (with an average token reduction of 61.31%), but also improves overall accuracy (with an average increase of 1.43%).** The gains are particularly notable on low to medium difficulty levels (e.g., accuracy improvements of over 2.5% on Levels 1 to 3). Although there is a slight drop in accuracy at Level 4 (-1.12%), the method still achieves over 65% token reduction, indicating that Adaptive Think effectively optimizes reasoning efficiency even on more complex tasks.
>
> Overall, these results further confirm the adaptability and efficiency of our method across varying levels of question difficulty.
>
> > **Q2: Derivation of the Upper Bound for Information Bias (Eq. 2) Is Incomplete and Needs Clarification**
>
> We appreciate the reviewer’s question regarding the derivation of the upper bound. **This bound is based on the mutual information estimation presented in ref.[1]**. Specifically, we consider the deviation between the empirical mutual information estimator $\hat{I}_N(S,T)$ and the true mutual information $I(S,T)$.
>
> The empirical estimate is computed $N$ i.i.d. samples $\{(s_i,t_i)\}_{i=1}^N$ drawn from the joint distribution: $$\hat{I}_N(S,T)=\hat{H}_N(S)+\hat{H}_N(T)-\hat{H}_N(S,T),$$ where $\hat{H}_N(\cdot)$ are empirical entropy estimates (e.g., kernel-based method we used in our experiments).
>
> The deviation $|\hat{I}_N(S,T)-I(S,T)|$ arises due to:
> - **Bias**: The empirical estimator $\hat{I}_N(S,T)$ may be biased, especially for small $N$.
> - **Variance**: Random fluctuations in finite-sample estimates.
>
> Based on McDiarmid's inequality (a generalization of Hoeffding's inequality for functions with bounded differences), for any $\epsilon>0$: $$P\left(|\hat{I}_N-\mathbb{E}[\hat{I}_N]|\geq \epsilon\right)\leq 2\exp\left(-\frac{2N\epsilon^2}{c^2}\right),$$ where $C$ represents the sum of squared bounds on the differences (in this case, $C = \mathcal{O}(1)$). Setting the confidence probability $1-\delta$, we equate the right-hand side to $\delta$ and solve for $\epsilon$: $\epsilon\propto \sqrt{\frac{2\log (2/\delta)}{N}}$. Combining this with the bias of the expectation provided by Paninski, $\mathbb{E}[\hat{I}_N(S,T)]-I(S,T)=\mathcal{O}(1/N)$ (due to the bias of entropy estimation), the total deviation becomes: $$|\hat{I}_N(S,T)-I(S,T)|\leq \sqrt{\frac{2\log(2/\delta)}{N}}+\mathcal{O}\left(\frac{1}{N}\right).$$
>
> [1] Paninski, L. (2003). *Estimation of entropy and mutual information*. Neural Computation, 15(6), 1191–1253.
> ***
> We sincerely appreciate your thoughtful comments and suggestions. We are committed to continuously improving this work. Please do let us know if our response has sufficiently addressed your concerns.

---

> ### Author Response · Authors · 2025-08-06
>
> Dear Reviewer 5Tnb,
>
> Thanks again for your great efforts and constructive advice in reviewing this paper! As the discussion period progresses and your comments are extremely valuable to improving our work, we expect your feedback and thoughts on our reply. We put a significant effort into our response, with several new experiments and discussions. We really hope you'll consider our reply. We look forward to hearing from you, and we can further address unclear explanations and remaining concerns if any.
>
> Best regards, ﻿
>
> Authors

---

> > ### Comment · Reviewer_5Tnb · 2025-08-06
> >
> > Thank you for your response. My concerns have been addressed, and I will maintain my positive scores.

---

> > > ### Author Response · Authors · 2025-08-07
> > >
> > > Thank you very much for your valuable suggestions and feedback, which have further strengthened our work! We will incorporate them into the next revision of our manuscript.

---

### Author Response · Authors · 2025-08-09
**General Response to the Reviewers and AC**

We extend our sincere gratitude to all Reviewers and Area Chair for their constructive feedback, thoughtful questions, and valuable suggestions, which have helped refine and strengthen this work. This study presents a novel information-theoretic perspective on reasoning efficiency in large reasoning models (LRMs), introducing entropy-grounded metrics to analyze overthinking and proposing a training-free Adaptive Think strategy to dynamically reduce reasoning length while preserving accuracy. Our findings advance both theoretical understanding and practical optimization of LRM inference.

We appreciate reviewers’ recognition of our contributions:
- Introduced two principled, information-theoretic metrics—InfoBias and InfoGain—to quantify semantic deviation and stepwise uncertainty reduction in reasoning chains, enabling rigorous efficiency analysis. Revealed key inefficiency patterns in long CoT reasoning, such as redundancy, semantic drift, and diminished marginal gains. (`5Tnb`, `uc9j`, `3P8b`, `SvkP`)
- Proposed Adaptive Think, an entropy-based, training-free test-time strategy that dynamically halts reasoning once sufficient confidence is reached, significantly reducing token usage while maintaining or improving accuracy across diverse reasoning tasks. (`5Tnb`, `uc9j`, `3P8b`)
- Presented a clear narrative structure and lucid presentation, making the problem definition, theoretical foundations, and experimental validations easy to follow and well-connected. (`uc9j`, `3P8b`, `SvkP`)

We have made every effort to address the concerns raised:
- **Model scale fairness** (`5Tnb`, `3P8b`): Included additional results on smaller reasoning models (e.g., DeepSeek-R1-Distill-Qwen-7B) and larger non-reasoning models to ensure fairer comparisons; results confirm Adaptive Think’s benefits across scales.

- **Dataset scope & statistical validity** (`5Tnb`, `SvkP`): Expanded to MATH500, which served two purposes: (1) in the analysis stage, controlling problem difficulty more precisely to enhance the robustness of InfoBias-based findings; and (2) in the evaluation stage, demonstrating that Adaptive Think consistently achieves efficiency gains and accuracy preservation across problems of varying difficulty levels.

- **Relation to existing methods** (`uc9j`): Clarified methodological distinctions from *Dynasor-CoT*, including termination criteria, reasoning granularity, and broader task adaptability, and highlighted our empirical advantages.

- **Theoretical grounding** (`3P8b`): Expanded explanation of how InfoBias and InfoGain map directly to Shannon entropy concepts, framing LRM reasoning as an information compression and transmission process.

- **Practical limitations** (`SvkP`): Quantified the computational overhead for both multiple-choice and free-form tasks, providing detailed analysis of its association to token savings, and discussed applicability boundaries for open-ended generation as well as potential extensions.

---

### Decision · Program_Chairs · 2025-09-17

**Decision:**

Accept (spotlight)

**Comment:**

The paper presents a novel information-theoretic perspective on reasoning efficiency in large reasoning models (LRMs), introducing entropy-grounded metrics to analyze overthinking and proposing a training-free Adaptive Think strategy to dynamically reduce reasoning length while preserving accuracy.

Strengths:

1. This paper redefines reasoning efficiency from an information-theoretic standpoint (entropy and mutual information), with two information theory-based metrics and employed them to analyze the LongCoT processes, finding some interesting phenomena that are in line with human intuition.
2. The paper proposes entropy-based Adaptive Think strategy, which is simple, straightforward, effective and training-free.
3. Experiments on different models and benchmarks illustrated that the proposed adaptive-think method shows better trade-off between effectiveness and efficiency.

Weakness:

1. Analysis only in GSM8K dataset (`5Tnb`, `uc9j`), the author address this during the rebuttal.
2. Concerns about generalizability and effectiveness of the proposed strategy across model scales, and dataset like open-end tasks (`3P8b`, `SvkP`).

All weakness has been addressed during rebuttal and discussion, and all reviewers agree to accept the paper. It is suggested to incorporate all revisions and suggestions in the next version.